# Ectoderm to mesoderm transition by down-regulation of actomyosin contractility

**Leily Kashkooli**[1,2◉], **David Rozema**[1,2◉], **Lina Espejo-Ramirez**[1], **Paul Lasko**[2], **François Fagotto**[1,2]*

1 CRBM, University of Montpellier and CNRS, Montpellier, France, 2 Department of Biology, McGill University, Montreal, Quebec, Canada

◉ These authors contributed equally to this work.
* francois.fagotto@crbm.cnrs.fr

## Abstract

Collective migration of cohesive tissues is a fundamental process in morphogenesis and is particularly well illustrated during gastrulation by the rapid and massive internalization of the mesoderm, which contrasts with the much more modest movements of the ectoderm. In the *Xenopus* embryo, the differences in morphogenetic capabilities of ectoderm and mesoderm can be connected to the intrinsic motility of individual cells, very low for ectoderm, high for mesoderm. Surprisingly, we find that these seemingly deep differences can be accounted for simply by differences in Rho-kinases (Rock)-dependent actomyosin contractility. We show that Rock inhibition is sufficient to rapidly unleash motility in the ectoderm and confer it with mesoderm-like properties. In the mesoderm, this motility is dependent on two negative regulators of RhoA, the small GTPase Rnd1 and the RhoGAP Shirin/Dlc2/ArhGAP37. Both are absolutely essential for gastrulation. At the cellular and tissue level, the two regulators show overlapping yet distinct functions. They both contribute to decrease cortical tension and confer motility, but Shirin tends to increase tissue fluidity and stimulate dispersion, while Rnd1 tends to favor more compact collective migration. Thus, each is able to contribute to a specific property of the migratory behavior of the mesoderm. We propose that the "ectoderm to mesoderm transition" is a prototypic case of collective migration driven by a down-regulation of cellular tension, without the need for the complex changes traditionally associated with the epithelial-to-mesenchymal transition.

## Introduction

The ability of tissues to dynamically rearrange is at the core of animal morphogenesis. In some systems, this is primarily accomplished by epithelial morphogenesis requiring cell shape changes or planar cell rearrangements. However, other systems rely instead on migration of cell masses. Gastrulation in *Xenopus* is a prototypical example of such type of morphogenesis [1]. Here, massive, coordinated cell migration results in the animally positioned ectoderm engulfing the vegetal endoderm, with the equatorial mesoderm positioned between the two germ layers. One of the main actors of early gastrulation is the prechordal mesoderm (PCM) which involutes and migrates collectively on the blastocoel roof (BCR) using the ectodermal

**Funding:** This work was funded by a Labex EpiGenMed Montpellier Chair of Excellence awarded to FF, (https://www.epigenmed.fr/index.php/funding/chair-of-excellence), and by grants awarded to FF from the Agence Nationale de la Recherche (ANR-14-ACHN-0004–ICM, https://anr.fr/) and from the Assocation pour la Recherche sur le Cancer Fundation (ARC grant 206766, https://www.fondation-arc.org), as well as as grant MOP-130350 from the Canadian Institute of Health Research (https://cihr-irsc.gc.ca), awarded to PL and FF. The funders had no role in study design, data collection and analysis, decision to publish, or preparation of the manuscript.

**Competing interests:** The authors have declared that no competing interests exist.

**Abbreviations:** BCR, blastocoel roof; CM, chordamesoderm; EMT, epithelial-to-mesenchymal transition; FA, focal adhesion; FN, fibronectin; LEM, leading edge mesendoderm; MLC, myosin light chain; MO, morpholino antisense oligonucleotide; MPA, micropipette aspiration technique; MYPT, myosin light chain phosphatase; PCM, prechordal mesoderm; pMLC, phosphorylated MLC; qPCR, quantitative PCR; relT, relative contact tension; Rock, Rho-kinases; TST, tissue surface tension; wtShi, wild-type Shirin; YFP, yellow fluorescent protein.

cells and a thinly deposited fibronectin (FN) matrix as substrates [1,2]. As it does so, there is ongoing intercellular migration within the tissue that leads to extensive radial intercalations, resulting in progressive thinning of the tissue until eventually all cells contact the BCR.

The PCM, which originates from the ectoderm through an inductive process, exhibits high migratory activity that contrast with the nonmotile ectoderm from which it is derived. At a first glance, this behavior appears to be related to the classical epithelial-to-mesenchymal transition (EMT) observed for cells escaping solid tumors. However, the mesoderm cells move inside the embryo as a compact mass. Furthermore, the early *Xenopus* embryo is already multi-layered, and the mesoderm derives from the deep ectoderm layer, which does not display apical–basal polarity at the time of gastrulation, removing one of the principle hurdles that must be overcome during a classical EMT. Therefore, in this simple system, one can directly witness a tissue acquiring a migratory behavior without loss of cell–cell adhesion or changes in polarity. We propose that this process, which we name the "ectoderm to mesoderm transition," or "mesoderm transition" for short, constitutes a basal mode, which can teach us a great deal about the core cellular mechanisms that control tissue dynamicity.

The *Xenopus* embryo offers the unique possibility to easily dissect specific tissues, prepare explants and/or dissociate them into single cells, allowing the study of intrinsic cell and tissue properties in the absence of confounding influences of other surrounding embryonic structures. Importantly, the morphogenetic events occurring during *Xenopus* gastrulation are recapitulated in isolated explants, and furthermore, even individual dissociated cells have characteristics that clearly relate to the properties of the corresponding tissues: Ectoderm cells show higher cortical stiffness, higher cell–cell adhesion, and are largely immotile, while the softer mesoderm cells spread and migrate when laid on an FN substrate, similar to the mesoderm at the BCR [1,3–5]. Note that there are two other mesodermal populations, the more anterior leading edge mesendoderm (LEM) and the posterior chordamesoderm (CM). The LEM migrates along the BCR in front of the PCM, while the CM undergoes the particular process of convergent extension at later stages of development. However, in this study, we focus on the PCM, which for the sake of simplicity is here referred to as "mesoderm."

We have based this investigation of mesoderm transition on the hypothesis that the high cortical contractility of ectoderm cells may be prohibitive for motility and that its decrease may be a key step in the mesoderm transition. We show that inhibition of the Rho–Rho-kinases (Rock) pathway is sufficient to confer ectoderm cells with migratory properties, which is the most fundamental aspect of gastrulating cells. We identify two mesoderm-specific negative regulators of RhoA, Rnd1 and Shirin (also called Dlc2, Stard13, or AhrGAP37), as absolutely required for gastrulation and more specifically for proper mesoderm migration, as predicted from our initial hypothesis. Our analysis of the impact of these regulators at the cell and tissue level supports a model where Rnd1 and Shirin cooperate toward a general down-regulation of actomyosin contractility, allowing cells to become motile, but also have opposing activities, with Shirin negatively impacting cell–cell adhesion leading to dispersive migration, while Rnd1 is capable of maintaining it for efficient collective migration. These differing roles likely balance each other to produce the right physical properties for effective mesoderm involution and intercalation.

## Results

### Distinct characteristics of ectoderm and mesoderm at the cell level

We first studied ectoderm and mesoderm cells in vitro in order to firmly characterize their basic intrinsic properties. Dissociated cells from early gastrula tissues were plated on FN and imaged by live confocal microscopy. FN is the major extracellular matrix component in the

gastrula, where it forms a sparse network [6,7]. Accordingly, we used low levels of FN for all our assays. Ectoderm and mesoderm cells have radically different morphologies and behavior: Ectoderm cells typically remain round and produce large blebs (Fig 1A and 1F) [8], and they do not migrate (Fig 1G, S1 Movie) [4]. Note that the small apparent "speed" (Fig 1G) does not represent actual migration, but reflects wobbling, as cells are "shaken" by constant blebbing (S1 Movie). On the contrary, mesoderm cells spread, form multiple prominent protrusions (Fig 1B and 1F', S2 Movie), and migrate at high speed (Fig 1G) [4]. Single mesoderm cell migration typically has low persistence, with one of the extended lamellipodia rapidly commuted to the cell's tail (S1A Fig) [2]. As a consequence, protrusive and retracting structures can be considered as oscillating states, unlike the strongly polarized extensions of many classical mesenchymal cell types.

The organization of matrix adhesions, marked by vinculin and paxillin, completely accounted for the differences in morphology and behavior, as mesoderm cells displayed typical vinculin and paxillin-positive focal adhesions (FAs) (Fig 1B'). These FAs were rapidly remodeled during migration (Fig 1F'). Ectoderm cells showed a completely different organization, harboring a highly stereotypical ring-shaped vinculin and paxillin-rich structure (Fig 1A' and 1F). These rings were immobile (Fig 1F, S1 Movie). Note that mesoderm cells displayed a spectrum of protrusions, from large lamellipodia to thin extensions, which all showed vinculin and paxillin enriched structures (Fig 1D' and 1F'). For simplicity, we will refer here to all the vinculin-rich structures detected on the ventral cell surface as FAs. Note also that in all subsequent experiments, we only tracked vinculin. Its absence did not preclude the occurrence of vinculin-negative FAs, but vinculin recruitment is an established parameter reflecting the tension exerted on adhesive structures [9,10]. We quantified the fraction of Vinculin-Cherry detected on the ventral surface that concentrated at FAs in ectoderm and mesoderm cells. We verified that this fraction is independent of expression levels (S1B Fig). The peculiar ectodermal adhesive rings concentrated high amounts of vinculin (Fig 1E), suggesting that these cells were interacting rather strongly with the substrate. We thus compared adhesion to FN by a rotation assay (Fig 1H). Ectoderm cells adhered almost as efficiently as mesoderm cells. This important observation indicated that the known inability of ectoderm cells to spread and migrate on FN was not due, as one may have hypothesized, to lack of efficient cell-matrix adhesion, but rather to an intrinsic property to organize a different type of adhesive structure. We used the same adhesion assay to compare cadherin-based adhesion, replacing FN with recombinant C-cadherin extracellular domain as the adhesive substrate. Ectoderm cells showed significantly higher cadherin adhesion than mesoderm cells (Fig 1H), consistent with previous measurements [3,11]. However, the difference was relatively mild, an observation that became relevant later in this study.

The analysis of small groups of cells showed that the properties of single dissociated cells were directly reflected at the supracellular level, each cell type adopting a distinct, highly stereotypic organization (Fig 1C and 1D): Ectoderm cells formed compact groups; they still emitted blebs, but exclusively along the edge of the group (Fig 1C). Cells did form some protrusions that crawled under adjacent cells, but typically in an inwards orientation (Fig 1C, yellow concave arrows). The cell group shared a multicellular vinculin/paxillin ring constituted by the juxtaposition of partial rings formed by the individual cells (Fig 1C', arrowheads). On the contrary, mesoderm cells formed widely spread groups with numerous lamellipodia. Both peripheral and internal lamellipodia were oriented outwards (Fig 1D and 1D', white and yellow concave arrows). FAs were aligned along the outward direction of the expanding protrusions (arrowheads).

This characterization highlighted deep intrinsic differences between ectoderm and mesoderm cells, which resulted in very different morphologies and in distinct adhesive structures.

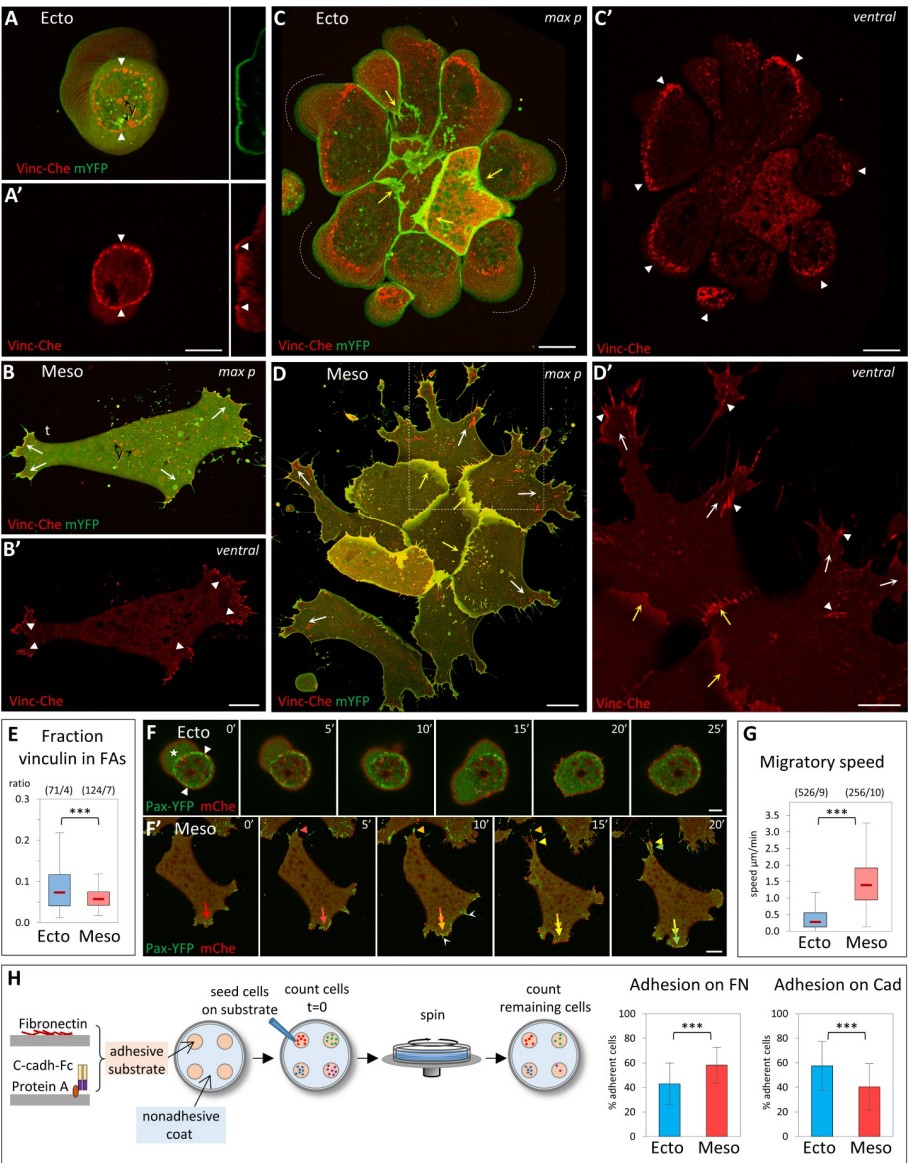

**Fig 1. Distinct properties of ectoderm and mesoderm at the cellular level.** (A–E) Organization of cell-matrix adhesive structures. Dissociated *Xenopus* ectoderm (A, C) and mesoderm (B, D) cells expressing Vinc-Che and mYFP were plated on FN, either as single cells (A, B) or as small groups (C, D) and imaged live by spinning disc confocal microscopy. y: autofluorescence of yolk platelets. Ventral: ventral z plane close to the glass. max p: Maximal z projection. (A) Ectoderm cells do not spread on FN, but adhere to it through a characteristic adhesive ring (A, A', filled arrowheads). They typically form blebs that are continuously pushed around the cell (dashed line with arrow). Right inserts: orthogonal view (orth) showing the cross-section of the membrane and of the vinculin ring (filled arrowheads). The dashed line underlines the bottom of the bleb. (B) Mesoderm cells spread on FN and extend multiple lamellipodia. They transiently polarize during their migration, with 1 protrusion becoming the tail (t); see also time lapse S1 Fig. They form vinculin-positive FAs (filled arrowheads), generally oriented in the direction of the protrusions (arrows). (C) Ectoderm cells form compact groups, with few protrusions in the center and numerous blebs at the periphery (dashed lines). External cells emit protrusions under the more central cells (yellow arrows). Individual cells build partial adhesive structures (filled arrowheads), which together form a supracellular ring. (D) Mesoderm cells form looser groups, each cell emitting multiple lamellipodia, most of them extending outwards (white and yellow arrows indicate peripheral and internal lamellipodia, respectively), with numerous FAs oriented radially (arrowheads). Panel D' is an enlargement of the boxed portion of panel D. Scale bars: A, C, D 10μm; B 20μm; D' 5μm. (E) Quantification of vinculin accumulation at FAs of isolated cells, expressed as Vinc-Che fluorescence concentrated in clusters divided by the total fluorescence along the ventral cortex. A color code is used throughout the figures, including blue for control ectoderm and red for control mesoderm. The box plots show the interquartile range (box limits), median (center line and corresponding value), and min and max values without outliers (whiskers). Statistical comparison using 2-sided

Student *t* test. For all experiments presented in this study, *P* values are indicated as follows: $^*P < 0.05$, $^{**}P < 0.01$, $^{***}P < 0.001$; NS, not significant. The same color code is also used to indicate statistical comparison between 1 condition and control ectoderm (blue) or control mesoderm (red). Other comparisons are indicated by black asterisks and brackets. The numbers in parentheses correspond to the number of cells/ number of experiments. Refer to S1 Data. (F) Single-cell motility. Frames of spinning disc confocal time-lapse movies. Cells expressed paxillin fused to YFP (Pax-YFP) and membrane Cherry. (F) Ectoderm cells are immobile, anchored by their stationary adhesive ring (arrowheads) and bleb (star). Scale bars: F 5 μm; F' 20 μm. (F') Mesoderm cells actively migrate, rapidly remodeling protrusions and FAs (red-yellow-green color-coded arrows and arrowheads indicate successive positions respectively of 1 extending lamellipodium and the retracting tail). White arrowheads: FAs at thin protrusions. (G) Quantification of single-cell migration. Refer to S1 Data. (H) Adhesion assay. Dissociated cells were plated on the adherent substrate, either FN or recombinant cadherin-Fc fusion protein, then subjected to rotation. Adhesion is expressed as the percentage of cells remaining adherent after rotation (see Materials and methods). The column plots show averages and standard deviation of 15 experiments, a total of approximately 5,000 cells/conditions. Statistical comparison on the % adherent cells/experiment, pairwise 2-sided Student *t* test. Refer to S1 Data. FA, focal adhesion; FN, fibronectin; mYFP, membrane-targeted yellow fluorescent protein; Vinc-Che, Vinculin-Cherry; YFP, yellow fluorescent protein.

These correlated well with their migratory capabilities, while differences in matrix and cell–cell adhesion were not as striking. Lastly, the properties observed for isolated cells readily translated into diametrically opposed collective organizations, compacted for ectoderm, expanded for mesoderm.

## Inhibition of Rock induces mesoderm-like spreading and migration of ectoderm cells

Ectoderm cells have intrinsically higher myosin-dependent cortical tension than mesoderm [3]. This high tension is reflected in cells plated on FN through their blebbing and by a stronger accumulation of cortical myosin light chain (MLC) (S2A–S2C Fig). We therefore hypothesized that differences in actomyosin contractility could be responsible for the distinct properties of ectoderm and mesoderm with respect to their spreading and migratory capabilities. Rock are important myosin activators. In both ectoderm and mesoderm cells, Rock1 and Rock2 are concentrated along the free cell cortex (S2D–S2K Fig, arrowheads), but present only at low levels at sites of cell-matrix and cell–cell adhesion (S2D–S2K Fig, arrows), consistent with a major role in controlling cortical tension.

We tested the effect of a short-term acute Rock inactivation on ectoderm cellular behavior using two specific chemical Rock inhibitors, Y27632 and H1152. The effect of these inhibitors on single ectoderm cells plated on FN was spectacular: Cells almost instantaneously stopped blebbing and within minutes started to spread, emit lamellipodia, and migrate (Fig 2A and 2B, S3 Movie). These changes were quantified by monitoring the increase in cell surface area (S3A and S3B Fig), the modification of cell morphology (Fig 2D), and by tracking migration (Fig 2E). In all these aspects, Rock inhibition appeared to be sufficient to induce a dramatic transformation of ectoderm cells into mesoderm-like cells, although the speed of migration remained significantly lower than that of mesoderm. Similarly, Rock inhibition caused groups of ectoderm cells (Fig 2C, S3C Fig, S4 Movie) to adopt the typical expanding configuration of mesoderm groups (compare to Fig 1D). The matrix adhesive structures were completely reorganized during this transition: The vinculin ring was disassembled, often starting asymmetrically, coinciding with extension of a protrusion and formation of classical FAs (S3C Fig for a small group, also seen in Fig 2A–2C). This observation further emphasized the congruence between single cell and collective behaviors. Importantly, the rapidity of the changes caused by the inhibitors (S3A and S3B Fig) clearly reflected a direct effect and excluded the involvement of transcriptional processes and changes in cell fate.

We also evaluated the effect of Rock inhibition on adhesion (S3D and S3E Fig). Rock inhibitors significantly increased adhesion of both ectoderm and mesoderm on FN. They also

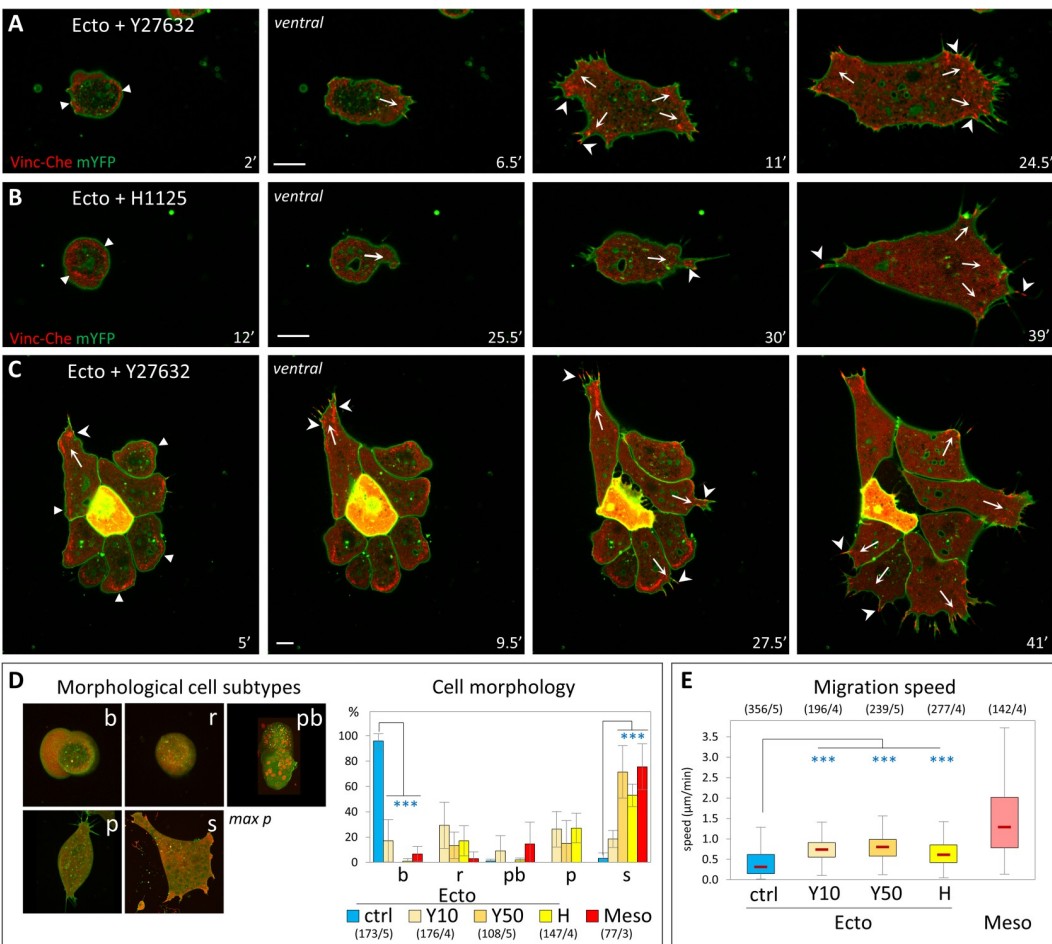

**Fig 2. Inhibition of Rock confers ectoderm cells with mesoderm-like properties.** (A-C) Induction of cell spreading and migration by Rock inhibition. (A–D) Confocal imaging of initiation of spreading and migration for single cells (A, B) and a small group of cells (C). Rock inhibitors, Y27632 (50 µM), and H1125 (1 µM) were added at time = 0'. Note that the onset of the transition is not synchronous. Arrows: nascent protrusions; filled arrowheads: ring-like adhesion; arrowheads: FAs. Scale bars: 10 µm. (D) Shift in cell morphology. Cells were classified in morphological subtypes: round and blebbing (b), round without blebs (r), polarized (p), and spread (s). In wild-type conditions, round cells are typically immotile, while polarized and spread cells migrate. A fifth category, named polarized with bleb (pb), includes cells with irregular morphology and blebs. The diagram shows the distribution of wild-type mesoderm and ectoderm cells, as well as of ectoderm cells treated for 50 minutes with 10 µM or 50 µM Y27632 (Y10 and Y50) or 1µM H1125 (H). For b and s categories, conditions were compared to control ectoderm by 1-way ANOVA followed by Tukey HSD post hoc test. Refer to S1 Data. (E) Migration speed of Rock-inhibited cells. Quantification as in Fig 1. Comparison to ectoderm control by 1-way ANOVA followed by Tukey HSD post hoc test. Refer to S1 Data. ANOVA, analysis of variance; FA, focal adhesion; HSD, honestly significant difference; mYFP, membrane-targeted YFP; Vinc-Che, Vinculin-Cherry.

increased adhesion on cadherin for mesoderm, without a detectable change for ectoderm. In stark contrast, the MLCK inhibitor ML7 potently inhibited adhesion of both tissues, on both FN and cadherin substrates. We concluded that both cell-matrix and cell–cell adhesions require MLCK activity, but not Rock activity. The latter, on the contrary, appears to act antagonistically to adhesion, which is precisely the expected impact of tension of the cell cortex, where Rock1/2 localize (S2D–S2K Fig). Together, these experiments support our initial hypothesis, pointing toward cortical Rock activity as a gatekeeper that prevents ectoderm from migrating. This model would predict that mesoderm cells should have acquired mechanisms to down-regulate cortical contractility in order to spread and migrate.

## Two Rho antagonists, Rnd1 and Shirin, are essential for mesoderm migratory and adhesive properties during gastrulation

The most parsimonious scenario that could account for the decreased myosin activity, lower cortical tension, and high motility of mesoderm was that this tissue expresses negative regulators of the Rho–Rock pathway. We searched through the *Xenopus laevis* developmental gene expression database (Xenbase, http://www.xenbase.org, [12]) for putative regulators expressed at the onset of gastrulation and determined by quantitative PCR (qPCR) their relative transcript levels in ectoderm and mesoderm. Two candidates, Rnd1 and Shirin, stood out as being significantly enriched in the mesoderm (Fig 3A). Rnd1 is a small GTPase that antagonizes RhoA through activation of ArhGAP35/p190B-RhoGAP and is implicated in the control of cell–cell adhesion [13]. Shirin/Dlc2/Stard13/ArhGAP37 is a RhoGAP, which has been associated with various functions, such as migration, adhesion, and cell division [14]. The potential role of these two regulators in the migratory properties of the mesoderm had not yet been addressed.

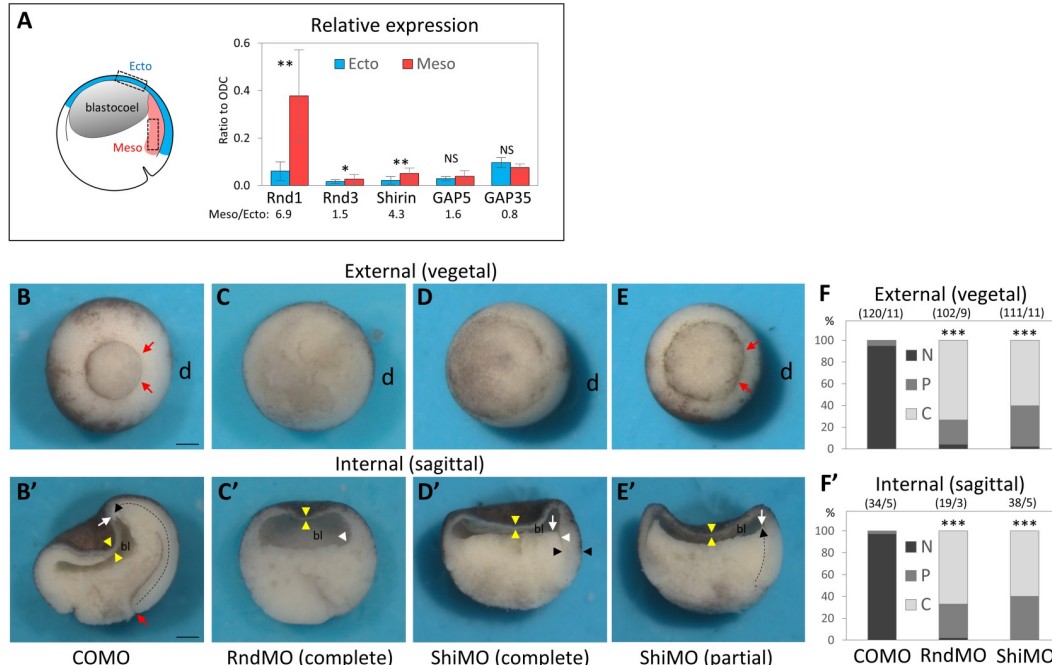

**Fig 3. Rnd1 and Shirin are essential for gastrulation.** (A) Rnd1 and Shirin expressions are enriched in the mesoderm. RT-qPCR from dissected tissue. mRNA levels in ectoderm and mesoderm, normalized to homogenously expressed ODC. Three to 6 experiments, pairwise 1-sided Student *t* test. Refer to S1 Data. (B–E) Whole embryo loss-of-function phenotypes: 4-cell stage embryos were injected in the dorsal side (d) with a control (COMO), Rnd1 (RndMO), or Shirin (ShiMO) morpholinos. Embryos were fixed and imaged at stage 11. (B-D) Examples of typical control mesoderm and RndMO and ShiMO phenotypes. (E) Example of a "partial" phenotype (here ShiMO). (B–E) External views from the vegetal pole. Red arrows point to the position of the dorsal blastopore lip of a control embryo, absent in RndMO (C) and ShiMO (D) embryos. (E) In the partial phenotype, the lip is present but the blastopore has remained widely open compared to control. In many embryos, the ventral blastopore is also affected, due to the diffusion of the morpholinos to the ventral blastomeres before complete separation after the 2nd cleavage. (B'–E') Sagittally bisected embryos. In a control embryo (B'), the extent of involution (dashed black arrow) can be seen by the position of the tip of the mesendoderm (white arrow) that has moved far away from the blastopore lip (red arrow). (C') RndMO embryo lacking any sign of involution. The white arrowhead points to the dorsal edge of the blastocoel cavity (bl), resembling that of a pregastrula embryo. (D') Characteristic ShiMO phenotype, with flat blastocoel floor (white arrow) and thicker non-involuted dorsal marginal zone (black arrowheads), both indicative of failed involution. (E') Partial involution (white arrow). Yellow arrowheads: thin BCR, indicative of ectoderm epiboly in all conditions. Scale bars: 200 μm. (F, F') Score of the penetrance of the gastrulation external and internal phenotype: N, normal embryo; P, partial inhibition, C, complete inhibition. Comparison by 1-way ANOVA followed by Tukey HSD post hoc test. Refer to S1 Data. ANOVA, analysis of variance; HSD, honestly significant difference; ODC, ornithine decarboxylase; RT-qPCR, reverse-transcription quantitative PCR.

Injection of specific morpholino antisense oligonucleotides (MOs) targeted against Rnd1 or Shirin mRNAs yielded severe gastrulation phenotypes, with virtually full penetrance (Fig 3B–3F). In both cases, the dorsal blastopore lip was strongly reduced or missing altogether (Fig 3C–3F). The internal morphology was similarly deeply affected, revealing a block of mesoderm involution (Fig 3C'–3F'). Importantly, this dramatic defect contrasted with the normal thinning of the ectodermal BCR, which indicated that epiboly, another key morphogenetic movement during gastrulation, proceeded normally. We went on to investigate the Rnd1 and Shirin loss-of-function phenotypes at the tissue and cellular level. We started with the analysis of single dissociated cells. Rnd1MO and ShiMO had drastic effects: Most injected mesoderm cells failed to spread on FN and often showed blebbing (Fig 4A–4C, S5 Movie, quantification in Fig 4D). Their migration was significantly decreased compared to control mesoderm, the effect being strongest for ShirinMO (Fig 4E). Specificity of the Rnd1 and Shirin MOs was demonstrated by rescue of spreading and migration upon expression of Rnd1 and Shirin fused to yellow fluorescent protein (YFP; S4A–S4D Fig). Moreover, spreading and migration were also rescued by Rock inhibition, demonstrating that indeed Rnd1 and Shirin act upstream of Rock (S4A–S4D Fig). Simultaneous depletion of Rnd1 and Shirin led to an even stronger phenotype (Fig 4D and 4E), with an almost complete loss of migration (Fig 4E). We also found that adhesion on FN and cadherin were both significantly impaired (Fig 4G and 4H).

Beyond these common effects, we observed differences between Rnd1MO and ShiMO cellular phenotypes. Rnd1MO cells almost completely lacked detectable vinculin-positive FAs (Fig 4B and 4F), while vinculin distribution in ShiMO cells was heterogeneous (Fig 4F): Some ShiMO cells still harbored classical FAs, others had none, and others started to assemble peripheral concentric FAs strikingly reminiscent of the rings observed in ectoderm cells (Fig 4C). A closer look at migration brought further interesting insights. So far, we had compiled the average migration speed of all cells, independently of their morphology (Fig 4E). In order to better understand the cause of the decreased migration, we analyzed the speed of each category of cells (S4E Fig). While the overwhelming majority of wild-type mesoderm cells had a spread morphology, other types could be found at low frequency, which allowed us to confirm that the morphology correlated with migration: Spread cells showed the highest speed, while round cells (with or without blebs) showed the lowest. Nevertheless, round and blebbing mesoderm cells were still faster than ectoderm cells (1 μm/min versus less than 0.3 μm/min), indicating that, even for this typical "immobile" morphology, mesoderm cells remained capable of some migration. RndMO mesoderm cells showed an identical profile to control mesoderm throughout all categories (S4E Fig). We could conclude that the lower average speed of RndMO cells directly reflected their switch from spread to round morphology (Fig 4D). The profile was different for ShiMO: We calculated that the migration speed was significantly decreased by knockdown of Shirin for all morphological categories (S4E Fig), implying that ShiMO, in addition to causing a shift in morphology, had also a separate impact on motility. These data argued for a differential role of the two regulators.

In summary, the specific activation of Rnd1 and Shirin expression in mesoderm cells appears to be absolutely required for mesoderm involution, controlling cell spreading, motility, and adhesion and accounting for the predicted pro-migratory effect of down-regulation of the Rho–Rock pathway. However, the loss-of-function phenotypes clearly differed in several aspects, indicating that Rnd1 and Shirin had distinct activities.

## Expression of Rnd1 and Shirin confer ectoderm with mesoderm-like migratory properties

Next, we tested the effect of overexpressing Rnd1 or Shirin in ectoderm cells, with the rationale that they may reproduce the transition toward a mesoderm-like phenotype observed upon

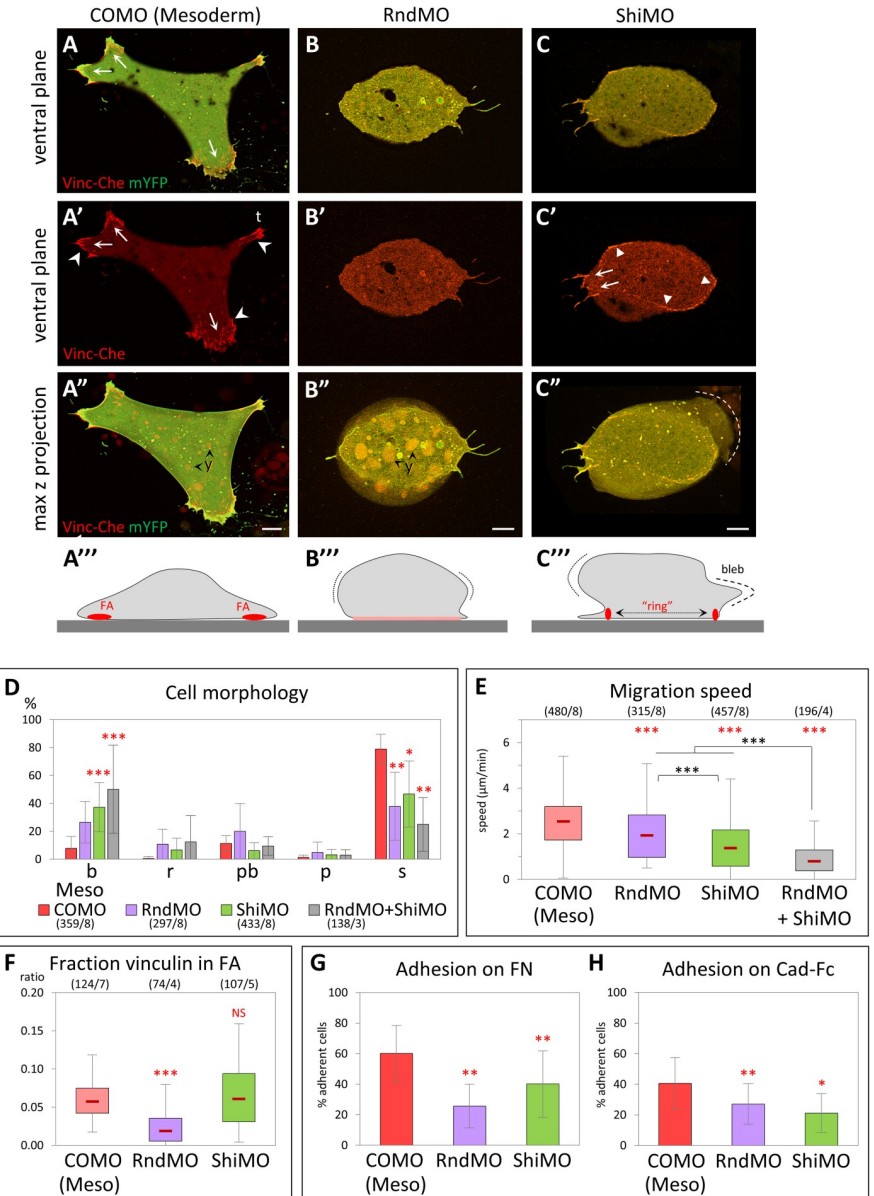

**Fig 4. Rnd1 and Shirin are essential for mesoderm spreading and migration.** (A–E) Loss-of-function cellular phenotypes. (A–C) Examples of control morpholino (COMO), RndMO, and ShiMO mesoderm cells, expressing Vinc-Che and mYFP, plated on FN. (A–C) Ventral z planes, merged channels; (A'–C') vinculin alone; (A''–C'') maximal z projections. (A'''–C''') Schematic diagrams summarizing the general cell morphology and adhesive structures. Protrusions are indicated by arrows, FAs by concave arrowheads, vinculin ring by filled arrowheads. Dotted lines highlight the max lateral extension of the cell mass. (A) Control spread mesoderm with large protrusions and numerous FAs. t, tail. (B) Typical RndMO cell displaying a bulging body (B'', dotted lines) and a small ventral surface with diffuse vinculin (B''', pink line). (C) Example of a bulky ShiMO cell with intermediate adhesive structures, including small FAs at short protrusions (arrows) and ectoderm-like partial ring encompassing most of the ventral surface (arrowheads). A bleb is visible in the max projection (C'', dashed line). y, yolk platelets. Scale bars 10 μm. (D) RndMO and ShiMO cells show a significant shift in morphology from spread to round and blebbing cell. Comparison for either of the 2 categories with corresponding COMO (red asterisks), 1-way ANOVA followed by Tukey HSD post hoc test. Refer to S1 Data. (E) Both RndMO and ShiMO inhibit cell migration. Gray asterisks: comparison with double injection RndMO + ShiMO, which significantly enhanced the migration phenotype. One-way ANOVA followed by Tukey HSD post hoc test. Refer to S1 Data. (F) Quantification of vinculin accumulation. Comparison to COMO using pairwise 2-sided Student *t* test. Refer to S1 Data. RndMO cells have little to no detectable vinculin-rich structures. ShiMO cells show high variability (see main text). (G, H) Inhibition of cell adhesion on FN and on cadherin substrates. Five experiments, a total of 375–740 cells/conditions for FN, >1,000 cells for cadherin. Statistical comparison on the % adherent cells/experiment, pairwise 2-sided Student *t* test. Refer to S1 Data. ANOVA, analysis of variance; FA, focal

adhesion; FN, fibronectin; HSD, honestly significant difference; mYFP, membrane-targeted YFP; Vinc-Che, Vinculin-Cherry.

Rock inhibition. Indeed, both Rnd1 and Shirin induced remarkable changes in ectoderm cells: The frequency of blebs was strongly decreased, and a significant number of cells spread on FN, elongated, and extended protrusions (Fig 5B–5E and 5G), and became motile (Fig 5H and 5I, S6 Movie). Thus, either of these components was indeed capable to drive ectoderm cells into a migratory mode.

However, we also observed clear differences in the effect of the two regulators: Shirin was extremely potent at inducing cell spreading and formation of protrusions (Fig 5D, 5E and 5G), while Rnd1-expressing cells remained more circular and formed more modest protrusions (Fig 5B, 5C and 5G; see S5A Fig for a detailed quantitative morphometric analysis). On the other hand, Rnd1 had a higher pro-migratory activity (Fig 5I, S5B Fig). Note that cells tended to round up again for high expression levels (S5A Fig). The effect was marginal for Rnd1 (Fig 5C), but strong for Shirin (Fig 5E, S5A''' Fig), which, at the same time, induced extension of multiple tentacular protrusions (Fig 5E, S5A Fig). This effect presumably resulted from excessive loss of contractility (see below) and may explain stalling of migration (last frames of Fig 5H'' and S6 Movie). In terms of vinculin localization, most Rnd1- and Shirin-expressing cells lacked ring structures, and some FA-like structures could be observed in cells expressing moderate levels of either regulator (Fig 5B and 5D, arrows). Neither Rnd1 nor Shirin expression led to detectable changes in adhesion on FN (Fig 5J), but Shirin significantly decreased cadherin adhesion (Fig 5K). In conclusion, these experiments showed that both Rnd1 and Shirin could induce spreading and migration, but each expressed this property in a slightly different manner, further supporting overlapping yet diverging activities.

## Impact of Rnd1 and Shirin expression on ectoderm organization and myosin activation

We also analyzed the effect of ectopic Rnd1 and Shirin on ectoderm tissue in situ, by performing immunofluorescence on cryosections of whole embryos (Fig 6, S6 Fig). We focused on two key components, i.e., levels of phosphorylated MLC (pMLC) at the cell cortex, and cadherin-based adhesive contacts, marked by β-catenin. Membrane and cortex largely overlapped at the resolution used in these experiments, and β-catenin labeling could be used to segment what we generically call the "cell periphery," from which we quantified the relative signals of both markers. The ectoderm and the involuted PCM of wild-type embryos were used as reference (Fig 6A and 6B). Consistent with previous reports [3,15], levels of β-catenin and pMLC levels were significantly lower in the mesoderm (75% and 50%, respectively). Expression of Rnd1 or Shirin caused a significant decrease in cortical pMLC levels (Fig 6C, 6E''' and 6F'''), consistent with their role as negative regulators of the Rho–myosin pathway. The negative effect of Rnd1 and Shirin on pMLC cortical levels was also observed in dissected ectoderm explants, which provide large homogenous fields of cells, particularly favorable for quantitative immunofluorescence (S6F–S6I Fig). It is important to highlight the fact that in whole embryos and tissues, the peripheral or "cortical" pMLC signal results from contributions of the cytoskeleton associated with cell–cell contacts, cell-matrix contacts (FN secreted by the ectoderm), and "free" edges. These multiple inputs cannot be dissected apart in this complex setting.

We also verified that Rnd1 and Shirin acted via Rock by western blot analysis of phosphorylation of the regulatory subunit of myosin light chain phosphatase (MYPT), a direct target of Rock1/2, in ectoderm explants. Expression of Rnd1 and Shirin significantly decreased levels of phosphorylated MYPT (S6H Fig).

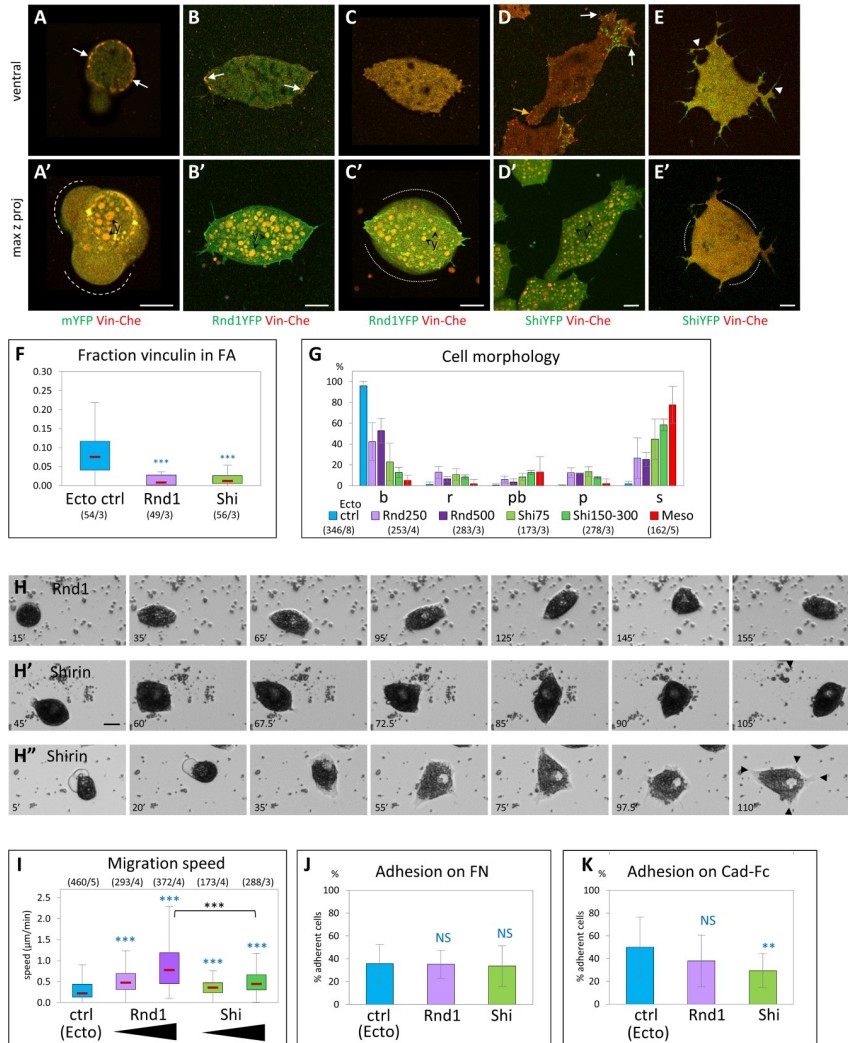

**Fig 5. Ectopic expression of Rnd1 or Shirin confers ectoderm with mesoderm-like morphological and migratory properties.** (A–G) Effect on cell morphology and vinculin distribution. (A–E) Examples of ectoderm cells co-expressing Vinc-Che and either mYFP (A, control ectoderm), Rnd1-YFP (B, C), or Shirin-YFP (Shi-YFP, D, E). (A) Typical control ectoderm cell, with its distinctive vinculin ring (arrows) and blebs (dashed lines). (B) Rnd1-expressing cells elongate, expand their ventral surface in contact with the substrate, but form only few vinculin-positive FA-like structures (arrows). (C) High Rnd1 expression: The ventral surface is expanded, but lacks vinculin FAs. Cells are bulkier (contours highlighted by dotted lines), although blebs are absent. (D) Shirin-expressing cells spread and form prominent lamellipodia with FAs (white arrows). The yellow arrow points the retracting tail. (E) High Shirin expression: Cells emit long and disorganized protrusions in all directions, but lack detectable FAs, and the cell body tends to round up (dotted lines). y, yolk platelets. Scale bars: 10 μm. (F) Quantification of vinculin accumulation. Consistent with the loss of the ring and the paucity of FAs, most of vinculin is homogeneously distributed on the ventral surface. Refer to S1 Data. (G) Distribution of morphological subtypes. Both Rnd1 and Shirin cause a strong shift toward spread cells. See S5A Fig for additional morphometric data. Refer to S1 Data. (H, I) Effect on cell migration and adhesion. (H) Frames from time-lapse movies. Examples of Rnd1- and Shirin-expressing ectoderm cells spreading and migrating. The cell in H" spreads extensively, ending with multiple protrusions (black arrowheads) and low motility. Scale bar: 20 μm. (I) Quantification of cell migration, as in Fig 2. Different levels of Rnd1 and Shirin expression were tested (250 and 500 pg mRNA for Rnd1, 75 and 150–300 pg for Shirin). Rnd1-expressing cells show higher migration than wild-type or Shi-expressing cells. Statistical comparisons: 1-way ANOVA followed by Tukey HSD post hoc test. Refer to S1 Data. (J, K) Quantification of cell adhesion on FN and on cadherin. Four to 5 experiments, >1,000 cells per condition. Statistical comparison on the % adherent cells/experiment, pairwise 1-sided Student $t$ test. Refer to S1 Data. ANOVA, analysis of variance; FA, focal adhesion; FN, fibronectin; HSD, honestly significant difference; mYFP, membrane-targeted YFP; Vinc-Che, Vinculin-Cherry.

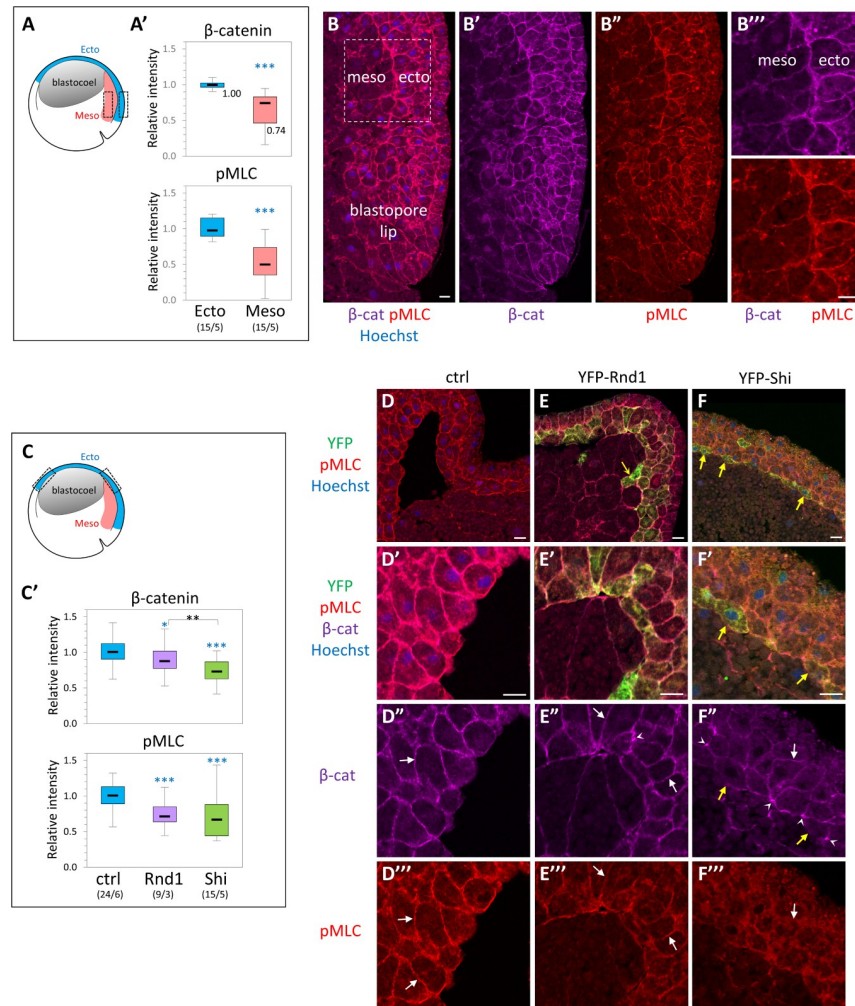

**Fig 6. Effect of ectopic expression of Rnd1 and Shirin on cell adhesive structures and cortical myosin.** β-catenin, used as general marker for cadherin-based cell adhesions, and pMLC were localized by immunofluorescence on cryosections of whole embryos at early gastrula stage. The fluorescence along the cell periphery, defined by the β-catenin signal, was quantified and expressed relative to the median intensity of control ectoderm. (A, B) Comparison of β-catenin and pMLC levels in the dorsal ectoderm and dorsal PCM of normal embryos. (A) Diagram of the embryo with boxes indicating the regions used for quantification. (A') Quantification. Numbers into brackets: number of embryos/number of experiments. Statistical comparison to ectoderm using 2-sided Student *t* test. (B) Example of dorsal region, immunolabeled for β-catenin (magenta) and pMLC (red). Nuclei were counterstained with Hoechst. (B''') Enlarged view of the region used for quantification. (C–G) Effect of Rnd1 and Shirin ectopic expression in the ectoderm. (C) Diagram indicating the regions of the ectoderm used for quantification. For consistency, all analyses were performed on the upper lateral region (both dorsal and ventral, indicated by dashed boxes in the diagram, because it constitutes a robust landmark where the inner ectoderm layer has a stereotyped organization. (C') Quantification. Statistical comparison using 2-sided, pairwise Student *t* test. Refer to S1 Data. (D-G) Examples of control (D), YFP-Rnd1-expressing (E), and YFP-Shirin-expressing (F) ectoderm, immunolabeled for the YFP-tag (green), β-catenin (far red, colored in magenta), and pMLC (red). Top panels present general views, and the other panels show enlarged portions of the inner ectoderm layer used for quantification. Note that the strong bending of the ectoderm layer is due to the partial collapse of the blastocoel cavity during fixation. White arrows point to plasma membranes marked by β-catenin (D', E', F') and to the corresponding pMLC signal (D", E", F"). Little to no pMLC enrichment is observed in Shirin-expressing cells (F"). Arrowheads in E' and F' point to concentrations of β-catenin, particularly frequent in Shirin-expressing ectoderm, and which contrast with the low membrane signal (arrow). Yellow arrow in E: Rnd1-expressing ectoderm cells that have penetrated into the mesoderm layer. Yellow arrows in F: ectoderm cells expressing particularly high levels of YFP-Shirin. PCM, prechordal mesoderm; pMLC, phosphorylated MLC.

Membrane β-catenin was also affected by expression of the two regulators. We observed a mild decrease for Rnd1 (Fig 6C and 6E"), but much stronger down-regulation for Shirin (Fig 6C and 6F"). Sparse punctate accumulations were observed, again more frequently for Shirin (arrowheads, Fig 6E" and 6F", S6A' Fig). The same uneven distribution was observed in ectoderm explants (S6H' and S6I' Fig). The strong effect of Shirin on β-catenin levels and distribution is fully consistent with the observed inhibition of cadherin-based adhesion by Shirin (Fig 5K).

The two regulators also strongly affected tissue organization (S6B–S6E Fig): In this region of wild-type embryos, cells of the deep ectoderm layer tend to orient roughly perpendicular to the inner surface (the BCR). This orientation was largely lost in Rnd1-expressing embryos (S6C and S6E Fig). Shirin expression led to a complete reorganization of the tissue. Cells became fusiform, aligning parallel to the ectoderm surface, forming a multilayered "mesenchymal-like" tissue (Fig 6F, S6A, S6D and S6E Fig), occasionally with large intercellular spaces (S6A Fig). In conclusion, these data demonstrated that Rnd1 and Shirin indeed negatively regulated cortical myosin activity, as well as cadherin-based adhesive structures and tissue organization, with the two latter features being most strongly affected by Shirin.

## Overlapping but distinct subcellular localization of Rnd1 and Shirin

In order to gain additional insights in Rnd1 and Shirin properties in these embryonic cells, we set to examine their subcellular localization. In the absence of adequate antibodies, we used the distribution of YFP fusion constructs as a proxy (Fig 7). In the *Xenopus* embryo, titration of injected mRNA allows the expression of low levels of these fluorescent constructs and to verify that subcellular patterns are reproducible even at the lowest detectable levels. For Shirin, because of the potent activity of RhoGAPs, even when expressed at low levels, we used a GAP-deficient R488A mutant (mShi) in order to visualize the subcellular distribution with minimal impact the cell phenotype.

Rnd1 was homogenously distributed along the cell cortex, both on the ventral side (Fig 7A') and at free cell edges (Fig 7A"), but significantly accumulated at cell–cell contacts (Fig 7A"). We measured a more than 2-fold enrichment at contacts, which is comparable to cadherin accumulation (Fig 7C). mShi also localized to the cell cortex, but, unlike Rnd1, it did not accumulate at cell–cell contacts (Fig 7B" and 7C). Instead, it formed prominent clusters on the ventral side of protrusions (Fig 7B, 7B', 7D and 7D'). We compared this ventral pattern with the localization of FAs, marked by Vinculin-Cherry. A large proportion of mShi clusters perfectly colocalized with vinculin-positive FAs (Fig 7D, quantification in Fig 7H). We also examined the localization of wild-type Shirin (wtShi), which, when expressed at low levels, did not induce overt changes in the morphology of mesoderm cells. wtShi localization was very similar to mShi, i.e., cortical with prominent accumulation of clusters in the protrusions (Fig 7E). However, wtShi had a strong impact on vinculin-positive FAs, which were largely excluded from the Shirin-rich regions and confined to the edge of the protrusions (Fig 7E). wtShi clusters showed limited overlap with FAs (Fig 7E, white and yellow concave arrowheads, quantification in Fig 7H). We also examined the protrusions induced by Shirin ectopic expression in ectoderm cells. The organization of these protrusions was similar to that of mesoderm protrusions, with accumulation of clustered Shirin and confinement of vinculin to the periphery (Fig 7F). Consistently, calculated colocalization was relatively low (Fig 7H). Interestingly, in ectoderm cells that had undergone an incomplete transition to a mesoderm-like phenotype (Fig 7G), Shirin lined the inner side of the remnants of the vinculin ring (white and orange arrowheads). One could conclude that Shirin is not only preferentially localized to protrusions, but more specifically targeted to FAs. While inactive Shirin accumulates at these structures,

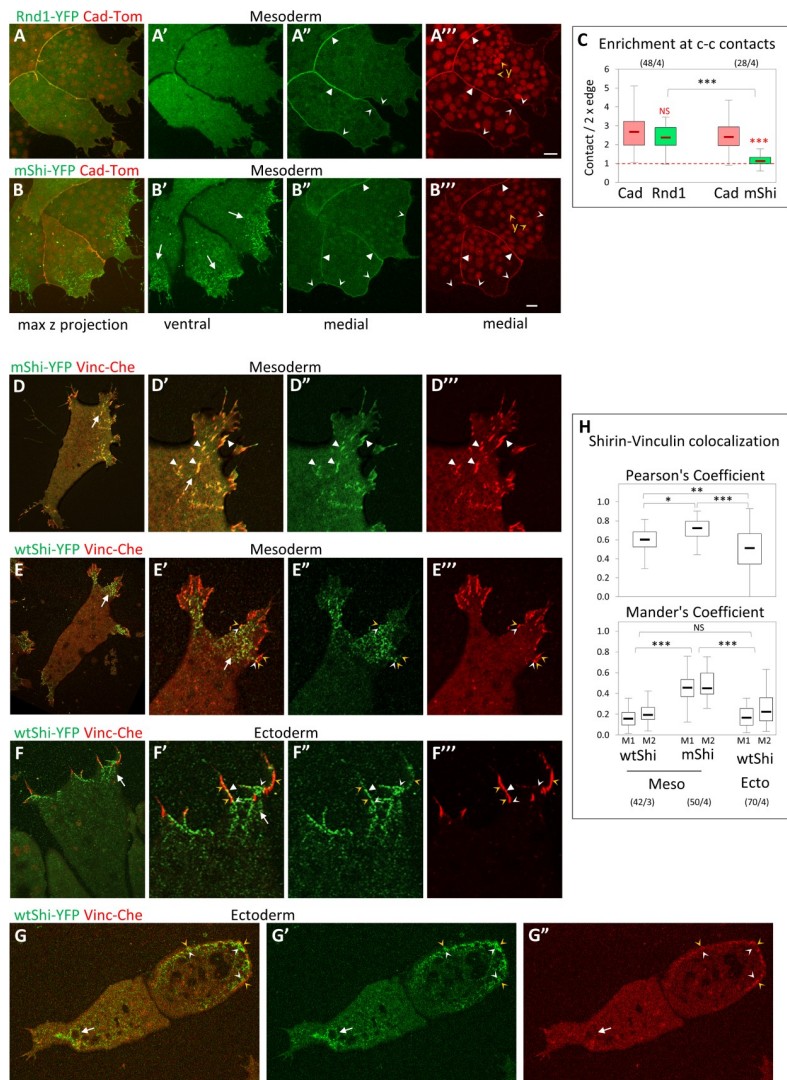

**Fig 7. Differential subcellular distribution of Rnd1 and Shirin.** (A, B) General distribution of Rnd1 and Shirin in mesoderm cells. (A, B) Live confocal microscope images of groups of mesoderm cells co-expressing cadherin-dTomato (Cad-Tom) and either Rnd1-YFP or GAP-deficient mutant ShirinR488A-YFP (mShi-YFP). Both Rnd1 and mShi localized to the cell cortex (concave arrowheads). On the ventral side, mShi was concentrated at protrusions (B', arrows, see D–H), while Rnd1 was always homogenously distributed (example in A'). Rnd1, but not mShi, is concentrated at cell–cell contacts (filled arrowheads). y, yolk platelets. (C) Quantification of Rnd1 and Shirin at cell–cell contacts, expressed as ratio of the signal intensity at cell–cell contacts divided by twice the signal along free cell edges. Rnd1 is enriched more than 2-fold at contacts, similar to cadherin. mShi is distributed homogenously along the cell periphery. Comparison Rnd1/mShi to cadherin (red) or mShi to Rnd1 (black) using 2-sided Student *t* test. Refer to S1 Data. (D–H) Shirin localization at the ventral surface. (D, E) Ventral surface of mesoderm cells co-expressing either mShi (D) or wtShi (E) together with Vinc-Che. (D, E) General view; D'–D''', E'–E''') enlargements of protrusions. mShi extensively colocalizes with vinculin at FAs (white arrowheads). (E) wtShi clusters are present throughout the ventral side of protrusions (arrow). Vinculin-positive FAs are largely confined to the periphery, only partially overlapping with wtShi clusters (orange arrowheads for vinculin, white concave arrowheads for wtShi). (F, G) Ectopic wtShi in ectoderm cells. (F) Detail of a protrusion of a fully spread cell. Similar to mesoderm, the center of the protrusion is occupied by clusters of wtShi and devoid of FAs (arrow). Small FAs are located at the periphery, close to Shirin clusters (orange and white concave arrowheads), but rarely colocalizing (white filled arrowhead). (G) Incompletely spread wtShi-expressing ectoderm cells. The left cell has lost its vinculin ring, and a wtShi-enriched protrusion is forming (arrow). The right cell still shows a weak ring lined in the inside by wtShi clusters (orange and white concave arrowheads). Scale bars: 10 μm. (H) Quantification of Shirin and Vinculin co-localization, expressed by the general Pearson's coefficient, as well as by Mander's coefficients, which indicates the portion of Shirin that overlap with Vinculin (M1) and the converse portion of Vinculin that overlap with Shirin (M2). Statistical comparison using 1-way ANOVA followed by Tukey HSD post hoc test. Refer to S1 Data. ANOVA, analysis of variance; FA, focal adhesion; HSD, honestly significant difference; Vinc-Che, Vinculin-Cherry; wtShi, wild-type Shirin.

expression of wtShi appears to "clear" vinculin from protrusions, consistent with its reported function in FA disassembly [14]. However, the presence of numerous FAs in non-manipulated mesoderm cells indicates that the normal function of endogenous Shirin is to moderate rather than to remove FAs altogether.

Most strikingly, the sites of Rnd1 and Shirin enrichment, respectively, at cell–cell contacts and in ventral protrusions, coincided with the two prominent regions where Rock1/2 were at their lowest level (S2 Fig). Together with the functional data, these observations suggested that both regulators contributed to the down-regulation of Rock-dependent cortical tension along free cell edges, while their complementary specific enrichments fulfill distinct functions: The ventral pool of Shirin would promote lamellipodium extension and keep tension at FAs under control at the cell-matrix interface, while Rnd1 would down-regulate tension at cell–cell contacts and help maintain cell–cell adhesion.

## Rnd1 and Shirin modulate cell surface tension and adhesiveness

To dissect the effect of Rnd1 and Shirin on cortical contractility and adhesiveness, we analyzed isolated cell doublets. In this simple system, the geometry of contact vertices directly reflects the balance of the forces exerted along the 3 interfaces, i.e., the cortical tensions along free edges ($Ct_A$ and $Ct_B$) and the contact tension $T_{AB}$ (Fig 8A) [3,16,17]. $T_{AB}$ is the sum of the two cortical tensions along the contact interface ($Ct_A$' and $Ct_B$', which are lower than $Ct_A$ and $Ct_B$) and of the negative contribution due to cell–cell adhesion (see S1 Appendix). Heterotypic doublets made of a wild-type ectoderm cell and a cell expressing Rnd1 or Shirin tended to be asymmetric, reflecting differences in their cortical tension (Fig 8B–8I). The asymmetry was particularly strong for Shirin: The heterotypic interface was systematically concave, with the Shirin-expressing cell engulfing the wild-type cell to various degrees (Fig 8H and 8I). We calculated that Ct was decreased about 2-fold in Shirin-expressing cells (Fig 8J). Rnd1 caused a more modest but significant reduction of about 10%. As comparison, we had previously shown that mesoderm cortical tension was about 2- to 3-fold lower than ectoderm [3]. Doublet geometry also allowed us to compare the relative contact tension (relT), which was significantly decreased by both Rnd1 and Shirin (Fig 8K). Since cell–cell adhesion is largely dictated by the reduction of cortical tension along the contacts [17–19], this reduction can be used to express a relative "adhesiveness", α, an absolute value that stands from 0 (no adhesion) to 1 (maximal adhesion) [19] (see S1 Appendix). Interestingly, α significantly decreased upon Shirin expression, but not in Rnd1 expression (Fig 8L). In summary, these measurements confirmed that both Rnd1 and Shirin repressed cortical tension, although to different extents. The stronger effect of Shirin explained why Shirin-expressing cells spread at higher frequency and more extensively (Fig 5, S5A Fig). These cell doublet experiments also provided key information about the differential impact of Rnd1 and Shirin on cell–cell adhesion. For Rnd1, the modest decrease in Ct was compensated by the parallel decrease in contact tension T, and adhesiveness was maintained. The balance between T and Ct was less favorable to adhesion in the case of Shirin, resulting in decreased adhesiveness. These biophysical measurements of adhesiveness were fully consistent with our adhesion assay, which showed that cadherin adhesion was significantly weakened by Shirin, but not by Rnd1 (Fig 5K), as well as with the stronger effect of Shirin on β-catenin, pMLC, and tissue organization observed in whole embryos (Fig 6).

## Rho/Rock regulation affects collective migration of ectoderm and mesoderm tissue explants

We extended our analysis to tissue-scale dynamics by investigating collective cell migration. For this purpose, we dissected tissue explants, let them heal for about 45 minutes until they

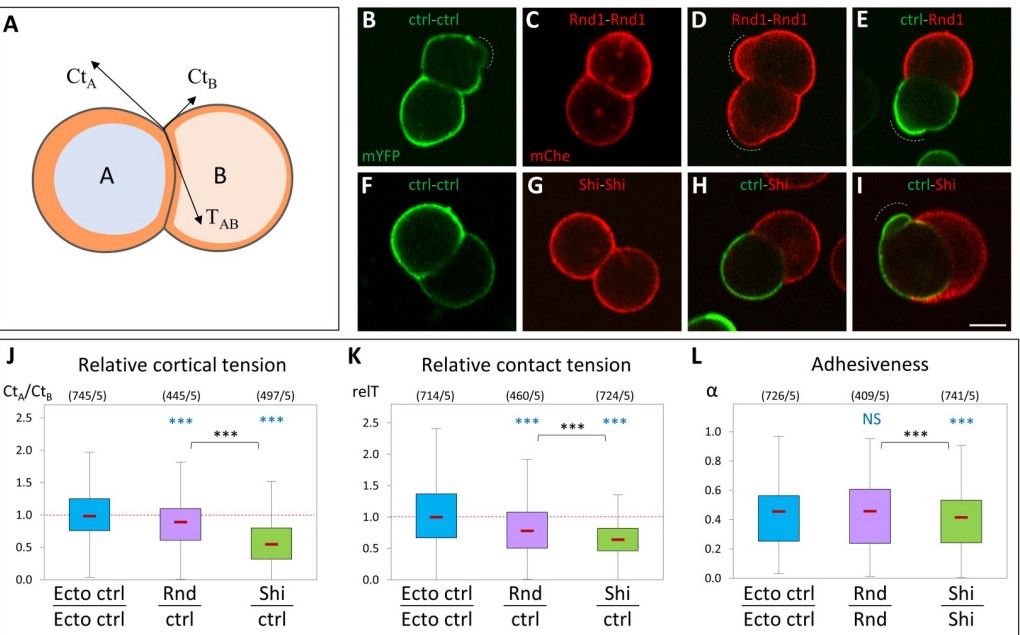

**Fig 8. Ectopic expression of Rnd1 or Shirin modulates ectoderm cortical tension and adhesiveness.** (A) Diagram of an asymmetrical cell doublet, representing the balance between cortical tensions at free edges $Ct_A$, $Ct_B$, and contact tension $T_{AB}$. The orange layer symbolizes the actomyosin cortex. The curved cell–cell interface reflects unequal $Ct_A$ and $Ct_B$ tensions. (B–I) Examples of homotypic and heterotypic doublets, imaged by live confocal microscopy. Doublets were made by combining dissociated control ectoderm cells expressing mYFP (ctrl) and either Rnd1 or Shirin-expressing cells markerd with mCherry. Wild-type and Rnd1-expressing cells often displayed blebs (dashed lines). Curved interfaces indicative of tensile differences were observed for all combinations, including for homotypic doublets (e.g., panel F), but were most systematically found for heterotypic ctrl-Shirin doublets (H, I). Scale bar: 20 μm. (J–L) Relative tension measurements based on the geometry at cell vertices (see S1 Appendix). (J) Relative cortical tension between Rnd1 or Shirin-expressing cells and control ectoderm cells calculated from the ratio $Ct_A/Ct_B$ of heterotypic doublets. The ratio for control homotypic doublets is provided for comparison. See S1 Appendix for complete measurements. Vertices flanked by a bleb (D and I) were omitted from calculations. (K) Relative strength of contact tension $T_{AB}$ at homotypic contacts, compared to control ectoderm-ectoderm T, the median of which was set arbitrarily at 1. See S1 Appendix for more details. (L) Relative adhesiveness α, calculated for homotypic doublets. Numbers in brackets: vertices/experiments. Statistical comparison using 1-way ANOVA followed by Tukey HSD post hoc test. Refer to S1 Data. ANOVA, analysis of variance; HSD, honestly significant difference.

formed a compact sphere, and then tested them for their ability to spread on FN for about 3 hours (Fig 9, S8 Fig). The behavior of the explants was quantitatively analyzed based on 3 parameters. (1) We first measured global explant spreading, expressed as relative total area expansion over time. (2) We further quantified the degree of dispersion during spreading using Delaunay triangulation of the nuclei. We determined the average triangle area between each trio of nuclei and calculated the relative ratio between the last point of the assay (170 minutes) and the beginning of spreading (30 minutes). Constant average area over time (ratio close to 1.0) meant that the tissue remained compact during spreading. A ratio greater than 1.0 was indicative of dispersion, a lower ratio suggested that the cells were further compacting. (3) We also measured the rate of intercalation by calculating the number of nuclei that were added to the basal cell layer (closest to the substrate).

In this assay, wild-type ectoderm explants did not spread on the substrate (Fig 9A, S7 Movie). Wild-type mesoderm explants quickly began to expand (Fig 9E, S11 Movie). Strikingly, however, their expansion was repeatedly interrupted by rapid, large-scale contractions (Fig 9E; red arrowheads, trace in S7B Fig). Mesoderm explants reached an apparent "steady-state" mode of alternating spreading and contraction, with an average maximal expansion 2-

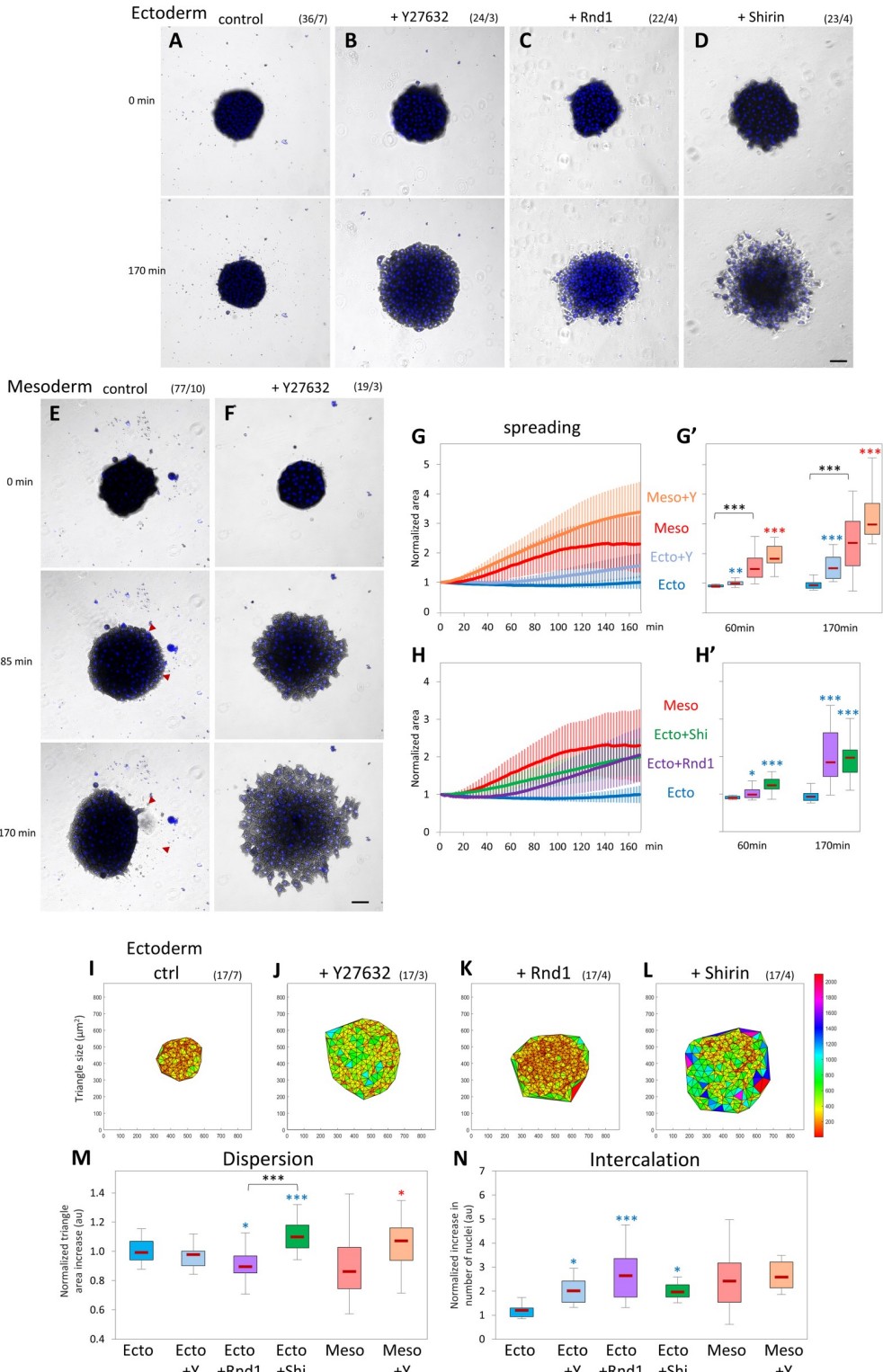

**Fig 9. Rho/Rock regulation affects collective migration of ectoderm and mesoderm tissue explants.** Tissue explants were laid on FN, and their spreading was imaged for 170 minutes. (A–D) Control ectoderm, ectoderm treated with 50-μM Y27632, and ectoderm expressing Rnd1 or Shirin. Numbers in brackets are number of explants and number of experiments. Scale bar: 100 μm. (E, F) Control mesoderm and mesoderm treated with Y27632. Red arrowheads in E indicate areas of large-scale retractions (compare 85 and 170 minutes). Scale bar: 100 μm. (G, H) Quantification of explant spreading. After segmentation, the area was calculated for the time course and normalized to the first time

point. Traces show average time course curves with SD for the various experimental conditions. (G', H') Corresponding relative spreading after 60 minutes and 170 minutes, chosen to represent an intermediate and advanced stage of the spreading process. Statistical analysis, 1-way ANOVA followed by Tukey HSD post hoc test. Refer to S1 Data. (I–M) Delaunay triangulation of nuclei and quantification of cell dispersion. (I–L) Representative maps of triangulated nuclei after 170 minutes of imaging. X and Y labels mark the coordinates in μm, and the color-coded scale bar indicates the area of the triangles in μm². (M) Quantification of the relative change in triangle size over time calculated by dividing the average triangle area at 170 minutes by that at 30 minutes. The 30-minute time point was chosen as it corresponds to the stage when explant had adhered to the substrate and started to spread. (N) Quantification of intercalation calculated by dividing number of nuclei at the ventral surface at 170 minutes by the number at 30 minutes. Statistical comparisons: 1-way ANOVA followed by Tukey HSD post hoc test. Refer to S1 Data. ANOVA, analysis of variance; FN, fibronectin; HSD, honestly significant difference; Rock, Rho-kinases; SD, standard deviation.

to 2.5-fold their initial size (Fig 9G and 9G'). These irregular phases of retraction explain the broad distribution of dispersion and intercalation data for this tissue (Fig 9M and 9N). This behavior suggested that mesoderm spreading was limited by internal tension. Consistently, the treatment of mesoderm with Y27632 completely abolished the retraction phases, leading to a smooth and broader expansion (Fig 9F, 9G and 9G', S12 Movie), which involved intense inter-calation (Fig 9N) but tended to be slightly dispersive (Fig 9M). Y27632 treatment also induced spreading of ectoderm explants (Fig 9B, 9G and 9G', S8 Movie). Interestingly, Y27632-treated ectoderm remained highly cohesive while capable of active intercalation (Fig 9O and 9P). Note that, at later time points (> 2 hours), a small proportion of wild-type ectoderm explants also started to spread (Fig 9G and 9G', S7A Fig), consistent with rare cases of spreading and migra-tion of single ectoderm cells (Fig 2J). The behavior of ectoderm and mesoderm tissues and the response to Rock inhibition were highly reminiscent of the behavior of single cells (Fig 2), emphasizing the connection between the cell autonomous characteristics and the tissue properties.

Expression of either Rnd1 or Shirin induced extensive spreading of the ectoderm explants (Fig 9C, 9D, 9H and 9H'). However, each regulator caused a distinct mode of spreading. Rnd1-expressing explants, after a delay, rapidly spread and efficiently intercalated (Fig 9C, 9H and 9N, S9 Movie). Migration remained thoroughly cohesive, and in fact, cells further com-pacted over time (Fig 9K and 9M). On the contrary, explants expressing Shirin became looser as they spread and partly disintegrated with numerous single cells migrating individually (Fig 9D, S10 Movie). Consistently, Delaunay triangulation confirmed cell dispersion, while interca-lation was comparatively lower than for Rnd1 (Fig 9L, 9M and 9N). This mode of migration was consistent with the negative effect of Shirin on cell–cell adhesiveness (Figs 5K, 6C and 8L).

The trends observed in ectopically expressing ectoderm were partially mirrored by the behavior of depleted mesoderm explants. Note that in these MO-based, Delaunay triangula-tion appeared different, both due to the intrinsically larger size of mesoderm cells, but also because in the Rnd and Shirin morphant conditions, the mesoderm explants were both looser and less adherent to FN (see inhibition of adhesion of single cells to both cadherin and FN, Fig 4G and 4H). As a result, at the early time points, there were very few nuclei in the morphant explants at the basal cell layer, leading to large initial triangle areas (S7K Fig) and likely an underestimation of dispersion and overestimation of intercalation. Nonetheless, clear conclu-sions could be made from the comparison of these conditions.

Shirin MO strongly decreased spreading (S8D, S8E and S8E' Fig, S14 Movie). Interestingly, intercalation was left intact (S8M Fig), and the tissue was strongly compacted (S8I and S8L Fig). Characteristic phases of spreading and contraction were still observed, but the average maximal expansion was much less than that of control mesoderm (S8D, S8E and S8E' Fig, red arrowheads). This phenotype can be explained considering that spreading was decreased due to Shirin depletion, while intercalation continued under the main influence of Rnd1. As for

Rnd1MO, it caused two distinct phenotypes, depending on the embryo batch: About one-third of the Rnd1MO mesoderm explants failed to spread. A majority of explants, however, spread quite extensively in an unusual way (S8C, S8E and S8E' Fig, S13 Movie): After a slow initial phase, rapid expansion and intercalation coincided with loss of cohesion among cells as they dispersed on the matrix (S8H and S8L Fig). This behavior is clearly reminiscent of cell dissemination observed in Shirin-expressing ectoderm (Fig 9D, 9L and 9M), suggesting that, in the absence of Rnd1, mesoderm behavior was dictated by the dispersive activity of Shirin.

## Effect of Rock inhibition and Rho regulators on tissue physical properties

To study global effects on physical properties of these tissues, we performed stress relaxation experiments using the micropipette aspiration technique (MPA). While the actual properties of tissues are quite complex, they can be modeled as viscoelastic materials, where the "elastic" component corresponds to short-term tissue behavior (determined by cortical tension and cell viscoelasticity), while the viscous component reflects the ability of the cells to actively rearrange within the tissue. In an MPA experiment, the initial fast deformation phase is dominated by the short-term properties and the slower subsequent phase by the long-term properties ("viscosity"). When the pressure is reset to 0, the aspirated portion of the explant will retract due to tissue surface tension (TST), which tends to restore the original spherical shape [20,21]. Stiffness and viscosity of the tissue offer resistance to the retraction, determining again a fast and a slow response. This model enables estimation of both tissue viscosity and TST based on the slopes of the slow viscous phases of aspiration and retraction (see Materials and methods) [20,21]. In addition, we have also quantified the initial fast deformation, as an indicator of the short-term "stiffness" of the tissue.

We observed clear differences in the behavior of ectoderm and mesoderm explants (Fig 10A and 10B, S15 and S16 Movies). During the initial fast phase, mesoderm explants were aspirated significantly deeper in the pipette (Fig 10A', 10B' and 10D). Viscosity calculated from the slow phases was also significantly lower for the mesoderm (Fig 10E), while TST was only slightly weaker (Fig 10F), in agreement with previous estimates [16]. Thus, mesoderm appears to be softer and more fluid than ectoderm, but maintains nevertheless a relatively high global tension. Different manipulations of the ectoderm gave distinct phenotypes (examples in Fig 10C, quantification in Fig 10D–10I): Y27632 treatment strongly decreased both viscosity and TST of ectoderm, but did not impact on the initial fast aspiration phase. Shirin expression strongly impacted all parameters, indicating that the tissue had become softer, more fluid, and less cohesive. On the other hand, TST was the sole parameter significantly decreased by Rnd1, and stiffness and viscosity remained largely unaffected. These results are in agreement with the cell doublet measurements and with the explant spreading data. Altogether, our data show that while mesoderm properties can be approximated as the result of a global decrease in Rock-dependent contractility, they are best accounted for by distinct actions of Rnd1 and Shirin. The former mostly operates on TST by moderating cortical tension while preserving cell–cell adhesion, while the latter stimulates tissue fluidity and dispersion by dampening both cortical tension and adhesiveness.

## Discussion

Ectoderm and mesoderm cells show diametrically opposed organizations in terms of cytoskeletal organization and adhesive structures, which explains their distinct migratory capabilities at both the single cell and tissue level. Yet, we could surprisingly easily convert ectoderm into a migratory, mesoderm-like tissue, by simply tuning down contractility via the Rho–Rock pathway. In fact, even non-manipulated ectoderm is capable, at low frequency, of spontaneous

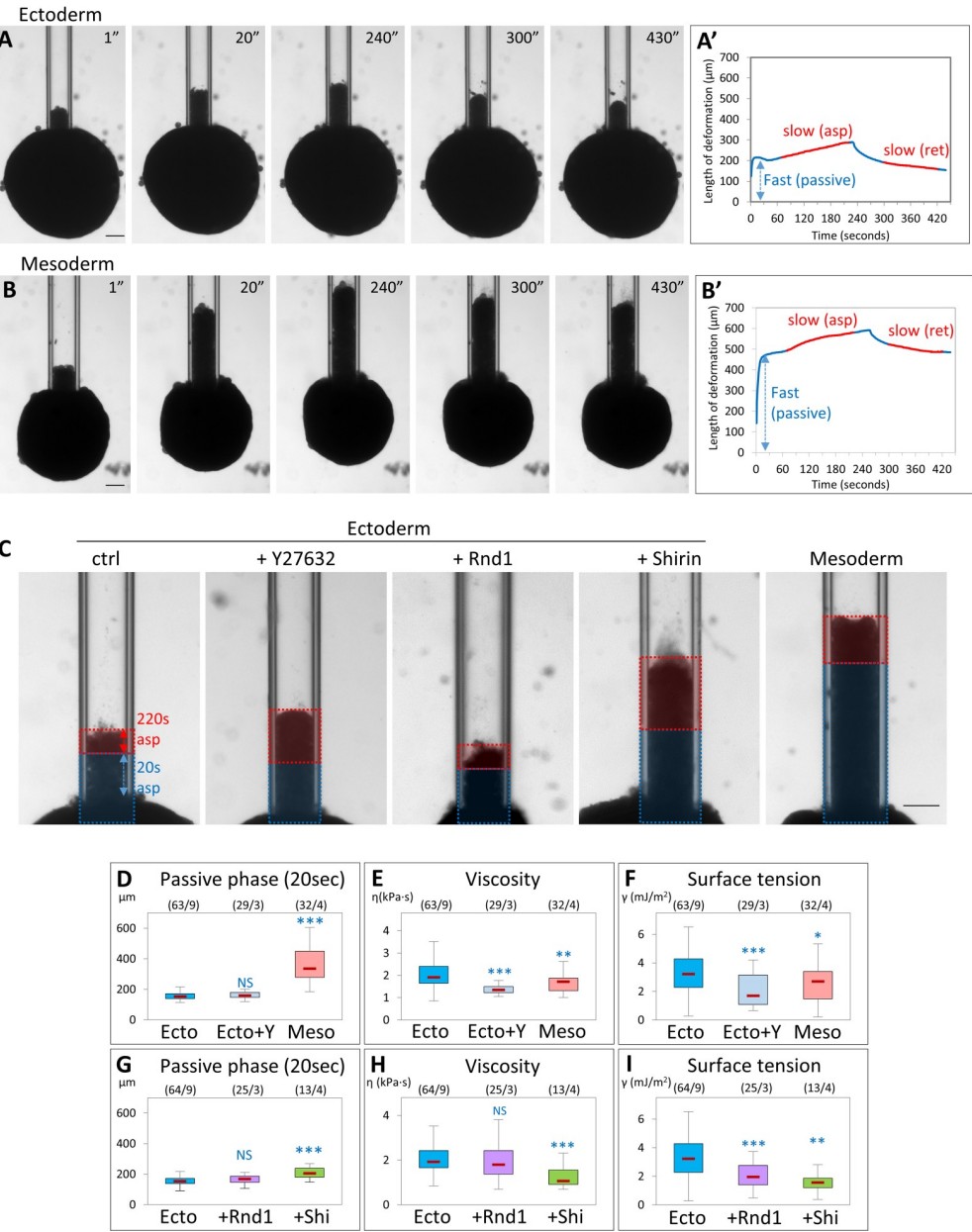

**Fig 10. ROCK inhibition and Rho regulators modulate tissue stiffness, viscosity, and surface tension.** Micropipette aspiration was used to measure physical properties of tissue explants. Explants were aspirated into the pipette at constant pressure, then pressure was reset to 0 to let the explant retract. (A, B) Examples of aspiration and retraction of control ectoderm and mesoderm explants. Aspiration pressure was 250 Pa. Pressure was released after 240 seconds. Scale bars: 100 μm. (A', B') Corresponding aspiration and release profiles. The blue double arrows indicate the extent of deformation of the tissue during the first 20 seconds, defined as the fast "passive" phase. The two slow, linear phases of aspiration and release, highlighted in red, were used to calculate viscosity and TST. Scale bars: 100 μm. (C) Examples of aspiration of control ectoderm, ectoderm treated with Y27632, or expressing Rnd1 or Shirin, and control mesoderm. Pressure was 250 Pa. Images display the frame corresponding to the deformation 220 seconds after the initiation of aspiration. The colored overlays indicate the distances of deformation during the first fast phase (20 seconds, blue) and during the subsequent slow phase (220 seconds, red). Scale bar: 100 μm. (D–I) Calculated parameters: (D, G) Length of deformation 20 seconds after initiation of aspiration, encompassing the initial passive phase. (E, H) Tissue viscosity calculated from the rates of aspiration and retraction (see Materials and methods). (F, I) TST. Numbers in brackets are number of explants and number of experiments. Statistical comparisons: 1-way ANOVA followed by Tukey HSD post hoc test. Refer to S1 Data. ANOVA, analysis of variance; HSD, honestly significant difference; Rock, Rho-kinases; TST, tissue surface tension.

spreading and migration (Fig 2J, S7 Fig). An important conclusion is that the ectoderm is not irreversibly locked into a nonmigratory configuration, but is actively maintained in a low dynamic state by its high contractility. Reciprocally, by targeting the mesoderm-specific Rho-negative regulators Rnd1 and Shirin, we could make mesoderm cells at least partly revert to a low-migratory, blebbing, ectoderm-like state. Quite remarkably, this reversion could go so far as to reproduce the characteristic concentric organization of adhesive structures (Fig 3D). These observations suggest that the seemingly deep morphological and behavioral dissimilarities between the two cell types derive from relatively simple molecular differences.

The transition from a static to migratory state is reminiscent of the maturation of pre-migratory precursors into migratory neural crest cells that occurs a few hours later [22]. In the neural crest model, the process is driven by a switch from E-cadherin to N-cadherin expression, leading to a shift from inwards to outwards protrusive activity, analogous to what we observe in gastrula tissues (Fig 1C and 1D) [22]. In this study, however, we find that the mesoderm transition seems to rely on a direct modulation of the cytoskeleton by expression of two negative regulators of RhoA. Most importantly, neural crest cells typically migrate in a relatively loose configuration, while the mesoderm remains coherent. Our various assays highlight the fascinating property of the mesoderm cells to be at the same time highly motile, which is observed both in isolation and in the tissue, and capable to be highly cohesive (Fig 9). This is amply confirmed by the fact that global tensile and viscous properties of this tissue are only marginally lower than those of the ectoderm (Fig 10), as previously reported by Winklbauer and colleagues [16,23]. Thus, this transition only involves a moderate shift in viscoelastic properties and clearly is not the result of a powerful fluidization. The necessity for the mesoderm to remain cohesive and tensile is obvious, as it must be able to generate and withstand the significant forces that are involved in gastrulation movements [24].

The apparent simplicity of this transition to high tissue motility, simply controlled by the regulation of cellular tension, contrast to the more complicated shift in parameters associated with EMT. It stands to reason that at this early stage of development, the "barrier to entry" of a migratory state is much smaller than that in more differentiated tissues. The same principle most likely applies to mesoderm involution in fish. Furthermore, we would like to propose that a similar down-regulation of contractility is likely a general mechanism to generate motility, which we expect to also be utilized in the EMT-like modes of gastrulation found in amniotes or in *Drosophila*.

Note that the viscosity and TST values obtained with the aspiration technique differ significantly from those reported by Winklbauer and colleagues [16,23]. This difference likely reflects the different techniques used and the different situations that were addressed: Previous work determined the properties of tissue explants through their global deformation under the sole influence of gravity, while we challenged the capacity of the tissues to resist local stress. Differences in "apparent" physical values have to be expected, considering that the cytoskeleton and cell adhesion are likely to behave differently under different stress conditions and that cells are capable of active reactions that can rapidly and deeply modify these structures and consequently tissue rheology. Therefore, these estimates must be considered as relative values, valid under specific experimental conditions. They are nevertheless highly informative about the properties of tissues and the influence of molecular manipulations. Along the same lines, the spreading assay (Fig 9) tests yet another situation, as spreading is controlled by the balance between the internal properties of the tissue (such as cell–cell adhesion, cortical tension, and intercellular motility) and the capacity of cells to spread, adhere, and migrate on extracellular matrix. We believe these various assays provide complementary approaches to unravel the mechanisms underlying morphogenetic processes.

This study implicates the Rnd1–Shirin pair as a key regulator of the ectoderm to mesoderm transition. Rnd1 and Shirin MO embryonic phenotypes are extremely strong and penetrant, demonstrating an absolute requirement of these molecules for mesoderm movements. At the tissue and cell level, Rnd1 and Shirin fulfill common as well as distinct complementary functions. They both promote mesoderm motility, as demonstrated by the decreased single-cell migration in Rnd1 MO and Shirin MO mesoderm, which is perfectly mirrored by induction of single-cell migration and of explant spreading in the gain-of-function experiments. This effect is in both cases related to their inhibitory activity toward the Rho–Rock pathway, resulting in significant reduction of cortical tension. Their activities, however, differ both quantitatively and qualitatively: In general, Shirin appears to be a more potent regulator, as observed for a variety of parameters examined in this study. Examples include stronger down-regulation of cortical tension, induction of cell spreading, decrease in β-catenin at cell contacts, and, to a lesser degree, in pMLC at the cortex. A likely explanation is that Shirin is a GAP, which has a direct catalytic activity on RhoA, while Rnd1 action is indirect. This apparent higher activity does not necessarily make Shirin "better" at all tasks. A good example is single-cell migration, for which Shirin is less efficient than Rnd1. This is not surprising, as adhesion and migration rely on a fine balance of myosin activity that depends not only on levels of activity, but also on additional parameters such as subcellular localization. These considerations also explain why global Rock inhibition, which efficiently stimulates spreading and migration in the ectoderm, falls short of reaching the migration capacity of mesoderm cells.

The two regulators also show clear qualitative differences in their action. Rnd1 appears specifically in charge of maintaining cell–cell contacts, while Shirin controls protrusive activity and FAs and negatively impacts cell–cell adhesion (although this may be an indirect effect of its powerful impact on the regulation of cortical tension). The complementarity of these two regulators in controlling collective migration was evident in the tissue spreading assay. In the hybrid phenotype of Rnd1 MO, the initial slower spreading was consistent with the contribution of Rnd1 in decreasing contractility and promoting motility. The subsequent emergence of a strong dispersive behavior revealed the underlying Shirin activity, which is otherwise counterbalanced by Rnd1 in the wild-type mesoderm. Consistently, ectopic expression of Shirin in the ectoderm caused a dispersive mode of spreading. Conversely, the prominent capacity of Rnd1 to stimulate intercalation, while imposing strong tissue coherence, was highlighted under conditions of Shirin depletion in the mesoderm and ectopic expression of Rnd1 in the ectoderm. Although these two regulators need to be further characterized, their doppelgänger nature is consistent with the overlapping yet partly complementary subcellular localizations. Thus, their global cortical pools are probably responsible for decreased cell cortical tension, spreading, and migration, while their sites of accumulation at protrusions for Shirin and at cell contacts for Rnd1 are consistent with their opposite effects on adhesiveness and tissue cohesion. The dual function of these regulators may also explain some less intuitive phenotypes, in particular the decreased cadherin adhesion for Shirin MO mesoderm cells, which most likely results from an imbalance in cellular tensions under these artificial conditions. Note that we also expect the input of additional components on the contractile and adhesive properties of these tissues, which remain to be identified.

The pro-migratory activity of both Rnd1 and Shirin uncovered here may seem unexpected, since Rnd1 and the Dlc1,2,3 family, to which Shirin belongs, are traditionally viewed as inhibitors of migration and as suppressors of invasion [14,25]. Rnd1 was also reported as an antiadhesive in *Xenopus* [13,26]. We also observed defects in adhesion and migration, but only in the case of cells expressing these regulators at high levels (Fig 5C and 5E, S5 Fig). This is to be expected considering the multiple effects of RhoA-dependent contractility and the intricacy of its regulation. The simplest interpretation is that overexpression of RhoA inhibitors brings

contractility below the basal level minimally required for adhesion and migration. Such considerations can readily explain why these molecules are found to have opposite effects depending on the cell type and the context [14,25]. The ability of Rnd1 to stimulate both migration and cohesion is reminiscent of the properties of EpCAM, a cell membrane protein that also acts as an indirect inhibitor of myosin contractility, although through a completely different pathway [27,28]. These types of regulators must share the ability to simultaneously repress global cortical tension, accounting for their pro-migratory activity and tension at cell–cell contacts, maintaining the proper force balance that insures tissue cohesiveness [16,17].

It is important to point out that some of the phenotypes observed at the tissue level could not be fully explained by the single cell experiments. For instance, Shirin expression induced extensive migration of ectoderm explants (Fig 9), but only modest migration of single cells (Fig 5). We similarly recognize that the in vitro analysis of isolated tissues may not reflect all the properties of these tissues in the in vivo context. Obviously, their morphogenesis in vivo is influenced by multiple factors, such as the geometry of the embryo, the forces exerted by the surrounding tissues, and the signals that they emit (e.g., the impact of ectodermal platelet-derived growth factor on mesoderm intercalation [29]). One thus must be cautious when extrapolating properties observed from a lower to an upper level of organization. Yet, studies on the *Xenopus* gastrula have amply demonstrated that explanted tissues retain many of their characteristic morphogenetic properties. Certain morphogenetic movements do appear to be at least partially tissue autonomous, including mesoderm involution and intercalation [30], vegetal rotation [31], or blastopore closure [32]. Furthermore, key cellular properties observed in the embryo and in isolated tissues are also retained in isolated cells. Directly relevant for this study, the differences in cell morphology and motility observed between subregions of the mesoderm (and endoderm) are preserved in isolated cells [4]. Even kinetics, which are expected to be most sensitive to changes in cellular and matrix environment, surprisingly remain within the physiological scale. Indeed, mesoderm cells migrate at a speed of approximately 1 μm/min during explant spreading (estimates based on experiments of Fig 9), which is not only close to the speed of single-cell migration (Fig 1), but also to the estimated speed in the embryo (approximately 2 μm/min R. Winklbauer, personal communication). Our results at the all 3 levels of organization provide a very coherent picture of the mechanism at the base of motility of the mesoderm, in particular in showing common and distinct activities of Rnd1 and Shirin.

Obviously, additional regulatory mechanisms are expected to fine tune the tissue properties in order to achieve the perfectly coordinated ballet of gastrulation movements. For instance, Rnd1 interactors were reported to modulate its function in the mesoderm [26,33]. We must stress, however, that both Shirin and Rnd1 are sufficient on their own to induce the distinct modes of migration described in this study, as shown unambiguously by the effect of their ectopic expression on single ectoderm cells and tissue explants. The cooperation of Rnd1 and Shirin/Dlc2 in enabling mesoderm involution provides an example of how different cytoskeletal regulators may be used to tune tissue behavior. It will be important to see if the same molecules or similar pairs of rivals contribute to other processes involving collective migration.

## Materials and methods

### Embryo preparation and injection

All plasmids are based on the pCS2+MTYFP vector [34]. Plasmids and morpholino oligonucleotides (Gene Tools, Philomath, United States of America) are listed in S1 and S2 Tables in the Supporting information section. mRNAs were synthesized according to the manufacturer's instructions (mMessage mMachine kit, Ambion, Austin, USA). MOs and mRNAs were injected animally in the 2 blastomeres of 2-cell stage embryos for ectoderm targeting, or

equatorially in the 2 dorsal blastomeres of 4-cell stage embryos for mesoderm targeting, at amounts listed in S1 and S2 Tables.

## Chemicals

Y27632, H1125, and ML7 were from Merck KGaA (Darmstadt, Germany) and Enzo Life Sciences France (Villeurbanne, France). Stock solutions of inhibitors were prepared in DMSO. They were used at a 1/1,000 or higher dilution. Equivalent dilutions of DMSO were added to control conditions and had no detectable effect on cell and tissue properties.

## Microdissections and cell dissociation

All dissected explants and cells were taken either from the inner layer of the ectodermal animal cap or from the anterior mesoderm at stage 10.5, except for the MO experiments, in which case the mesoderm was dissected from the dorsal lip at stage 10+, i.e., before involution. Dissections were performed in 1× MBSH (88 mM NaCl, 1 mM KCl, 2.4 mM NaHCO$_3$, 0.82 mM MgSO$_4$, 0.33 mM Ca(NO$_3$)$_2$, 0.33 mM CaCl$_2$, 10 mM Hepes, and 10 µg/ml Streptomycin and Penicillin, pH 7.4). Single cells were dissociated in alkaline buffer (88 mM NaCl, 1 mM KCl, and 10 mM NaHCO$_3$, pH = 9.5) [15]. All subsequent assays were performed in 0.5× MBSH buffer, at room temperature (23˚C).

## Western blots

Animal caps were dissected at stage 9 and allowed to heal on nonadhesive agarose-coated dishes until control embryos reached stage 10.25, after which protein was extracted. Rabbit anti-pMYPT1 Thr696 (Cell Signaling Technology, Danvers, USA) was used at 1:1,000, and anti-GAPDH FL-335 (Santa Cruz Biotech, Dallas, USA) was used at 1:4,000. A peroxidase conjugated donkey anti-rabbit secondary (Jackson Immuno Research, West Grove, USA) was used at a 1:4,000 dilution.

## Immunofluorescence

Whole embryos were fixed at stage 10.5 in 2% paraformaldehyde, 100 mM NaCal, and 100 mM HEPES-NaOH, pH 7.4 for 60 minutes, then permeabilized with 1% Triton X100 for 30 minutes and embedded in fish gelatine as previously described [35,36]. Ectoderm explants were dissecting at stage 9 (late blastula) and left to heal until control embryos at reached stage 10+, then fixed and processed as whole embryos. Cryosections were prepared and immunostained as described [35,36], except that Eriochrome counterstaining of the yolk was omitted in order to permit triple staining. Sections from multiple conditions, including a control condition, were collected on the same slide, in order to minimize immunostaining variability. Antibodies used were rabbit anti-β-catenin H102 (Santa Cruz Biotech) (1:200 dilution), mouse anti-phospho-myosin light chain 2 (Ser19) (Cell Signaling Technology) (1:200 dilution), and chicken anti-GFP (Merck, Darmstadt, Germany) (1:1,000 dilution). Secondary antibodies were coupled to Alexa488, 546, and 647 (Thermo Fisher Scientific, Waltham, USA). Nuclei were counterstained with Hoechst. Images were acquired on a SP5-SMD laser scanning confocal microscope (Leica Biosystems, Nussloch, Germany) with an oil immersion 20× objective (HC Plan Apo IMM 0.7NA).

## Live confocal microscopy

Glass bottom dishes (Cellvis, Mountain View, USA) were coated for 45 minutes with 10 µg/ml bovine FN (Merck) followed by blocking with 5 mg/ml bovine serum albumin. Dissociated

cells from embryos expressing various fluorescent fusion proteins were plated on the dish and imaged using a spinning disc confocal microscope (Dragonfly, Andor, Belfast, Northern Ireland), mounted with 2 EMCCD cameras (iXon888 Life Andor) for simultaneous dual color imaging, with a 60× objective (Apo lambda, 1.4 NA), and the Fusion acquisition software (Andor). Images were deconvoluted using Fusion software (Andor) and further analyzed using ImageJ (NIH, Bethesda, USA).

## Image analysis and quantification

All image quantification of confocal images was performed using ImageJ software.

Vinculin-Cherry enrichment was measured on maximal projections of two to three 0.25-µm thick z stacks encompassing the ventral cell surface. A mask was produced to extract the brighter signal of "clustered" Vinculin-Cherry corresponding to FAs. The total fluorescence intensity within this mask was divided by the total fluorescence intensity to the whole ventral surface of the cell, after background subtraction.

Quantification of colocalization between Shirin-YFP and Vinculin-Cherry at the ventral surface was performed using the JaCoP plug-in of Image J.

Relative cortical and contact enrichments of MLC-Cherry, Cadherin-dTomato, Rnd1-YFP, and Shirin-YFP were obtained by measuring the average fluorescence intensity of line scans manually drawn along free cell edges or along cell–cell contacts, as well as the intensity in the cytoplasm immediately adjacent to the cell periphery. After background subtraction, the "cortical" enrichment was calculated as cell edge (or cell contact)/cytoplasm.

Relative cell membrane/cortical enrichment of β-catenin and pMLC from immunofluorescence images were measured as follows: The β-catenin signal was used to produce a mask, which involved Gaussian filtering, 2 rounds background subtraction (global and local), thresholding to obtain a binary image, which was then skeletonized, and finally dilated to a thickness of 3 pixels (approximately 1.5 µm). The mask was used to extract the signal from both β-catenin and pMLC original images and measure the average intensity, to which the cytoplasmic background, obtained through a complementary mask, was subtracted. For embryo immunostaining, 3 fields of deep ectoderm cells were taken from each side (dorsal and ventral) of the embryo, on at least 2 different sections. For ectoderm explants, 2 large fields, together covering the majority of the section area, were imaged for each explant.

## Cell morphology, categories

The morphology analysis was performed from bright field time-lapse movies. The morphology of each single cell from a whole field was assessed at each time point of the migration assay and categorized as follows (examples in Fig 2J): round and blebbing (s), round not blebbing (r), polarized, i.e., elongated but still round-shaped or only partially spread (p), or spread (s). A fifth category included a special phenotype (polarized blebbing, pb), where cells were partially elongated, but had blebs and typically remained anchored to the substrate by 1 side of the cell. The distribution of morphologies presented in Figs 2D, 4D and 5G was expressed as the percentage of cells in these 5 categories observed at time 25'. The speed for each morphological category (S4 Fig) was calculated by extracting the average values for each segment of a track (within frames 10 and 40) during which the cell had adopted a particular morphology.

## Cell morphology, morphometry

The analysis was performed on stacks of live spinning confocal images. Two binary images were obtained, one from the ventral cell surface (closest to the glass), one from the maximal z

projection. Blebs were omitted from the segmentation. Absolute surface areas and circularity were obtained from the "measure object" function of ImageJ.

## Migration assay

Dissociated cells were plated on FN-coated glass bottom dishes and left to adhere for 45 to 60 minutes, then imaged every 2.5 minutes for 100 to 170 minutes using a bright field inverted Olympus IX83 microscope (Olympus, Tokyo, Japan) (10X UPFLN 0.3NA PH1 objective) and a scMOS ZYLA 4.2 MP camera (Andor). Chemical inhibitors were added after 4 frames (10 minutes) after the beginning of the time lapse. Addition of the inhibitor was set as time 0. The path of individual cells that did not establish contacts with neighboring cells were manually tracked using ImageJ software. Average speed corresponds to the average of the speeds calculated between each consecutive time point, within the window frames 10 to 40 (25 to 100 minutes).

## Adhesion assay

We used a modified assay based on Niessen and colleagues [37]. Moreover, 35-mm round dishes with a 20-mm diameter glass bottom (Cellvis) were freshly coated as follows: 1-mm diameter circles positioned near the edge of the cover glass, at 8.5 mm from the center, were coated with either 10 μg/ml FN (Merck) or with 100 μg/ml Protein A followed by 15 μg/μl recombinant C-cadherin extracellular domain fused to the Fc domain of human immunoglobulin G, produced and prepared as previously described [37]. Blocking buffer was as in [37]. Dissociated cells were laid in the coated circles and left to adhere for 45 minutes, and images with an inverted microscope mounted with a 5× objective were collected to determine the initial number of adherent cells. The dishes were then subjected to rotation (10 minutes at 180 rpm for FN and 25 minutes at 200 rpm for cadherin), and the fields were imaged a second time to determine the number of cells that had remained attached.

Calculation of relative tensions for cell doublets is presented in S1 Appendix.

## Tissue spreading assay

About 200 to 300-μm diameter explants were prepared by cutting pieces of dissected ectoderm or mesoderm tissues, which were left to heal and round up for 45 minutes on a nonadhesive agarose-coated dish. In cases of treatment with Y27632, the explants were incubated for an additional 45 minutes after healing. The explants were then transferred to FN-coated glass bottom dishes and imaged for 170 minutes every 2.5 minutes with a 10× objective as described for cell migration. Areas of explants were calculated at each time point using CellProfiler [38]. To measure cell dispersion during spreading, the XY coordinates of each nucleus was determined using CellProfiler. The coordinates were then used to perform Delaunay triangulation, followed by calculation of the area of each triangle using Matlab.

## Micropipette aspiration assay

MPA was used to measure the viscosity and surface tension of explants as previously described [20,21]. Custom-made pipettes with diameters of either 100 or 125 μm with a 15˚ bend (Sutter Instruments, Novato, USA) were passivated with BSA before being used to apply an aspiration pressure of 250 or 220 Pa (depending on the size of the pipette). The aspiration lasted 4 to 5 minutes, sufficient for the aspiration of the explant to reach a constant velocity; the pressure was then set to 0, and the explant was allowed to relax. The pressure was modulated using a Microfludic Flow Control System and the Maesflow software (Fluigent, Le Kremlin-Bicêtre,

France), and the pipettes were controlled using a PatchStar Micromanipulator and the Lin-Lab2 software (Scientifica, Uckfield, United Kingdom). The size of the deformation was automatically calculated using a custom ImageJ macro and used to calculate the rates of aspiration ($v_{Asp} = dL_{Asp}/dt$) and retraction ($v_{Ret} = dL_{Ret}/dt$) of the deformation, which were, in turn, used to calculate tissue viscosity and surface tension [21]. Briefly, viscosity $\eta = R_p\Delta P/3\pi(v_{Asp}+ v_{Ret})$, where $R_p$ is the radius of the pipette and $\Delta P$ is the applied pressure. Surface tension $\gamma = P_c/2^*$ $(1/R_p - 1/R_0)$, where $R_0$ is the radius of curvature of the explant, and $P_c$ is the pressure that when applied the length of the deformation is equal to $R_p$. It can also be calculated from $P_c = \Delta P$ $v_{Ret}/ (v_{Asp}+ v_{Ret})$. Images were acquired every 1 second using a brightfield Zeiss (Oberkochen, Germany) Axiovert 135TV microscope (5× Plan-Neofluar 0.15NA PH1) with a Retiga 2000R camera (QImaging, Surrey, Canada).

## Supporting information

**S1 Table. List of mRNA used in this study with injected amounts.**
(RTF)

**S2 Table. List of morpholinos with injected amounts.**
(PDF)

**S1 Fig.** (Related to Fig 1) (A) Mode of mesoderm locomotion. Consecutive frames from time lapse of mYFP labeled mesoderm cells migrating on FN. The behavior of the central cell is highlighted: The cell emits 1 or multiple protrusions (red arrows). One of the protrusions becomes a tail (yellow arrowhead) as the cell stretches toward another direction, and eventually retracts (red arrowheads). (B) Quantification of accumulation of Vinculin-Cherry in FAs: linearity between fluorescence levels in FA and total intensity (related to Fig 1A–1D). Because Vinculin-Cherry expression levels vary from cell to cell, quantification was performed for individual cells by measuring fluorescence in bright clusters (corresponding to FAs) and in the total ventral cell surface (pink on the diagram). The plot shows the average intensity of the ventral surface versus the average intensity in FAs for control mesoderm cells in 1 experiment, each dot corresponding to a single cell. It shows that accumulation at FAs is proportional to total expression levels over a wide range. Refer to S1 Data. Linearity was similarly verified for each experiment. FA, focal adhesion; FN, fibronectin; mYFP, membrane-targeted yellow fluorescent protein.
(PDF)

**S2 Fig. Localization of MLC and Rock (related to Fig 2).** (A–C) Differential MLC accumulation at the cell cortex. Ectoderm and mesoderm cells expressing MLC-Cherry (MLC-Che) and mYFP. (A) Ectoderm cells show strong accumulation around the cell body (arrows) and part of the blebs (arrowhead). (B) Mesoderm cells show irregular cortical MLC, mostly at the concave regions near or between protrusion. (C) Quantification of cortical MLC, expressed as the ratio of cortical /cytoplasmic fluorescence intensities. Blebs and protrusions were excluded from the measurements. Statistical comparison using 2-sided Student $t$ test. Refer to S1 Data. Scale bars: A' 5 μm, B' 10 μm, B" 5 μm. (D–K) Subcellular localization of Rock1-YFP and Rock2-YFP in ectoderm and mesoderm cells. Selected single planes from live confocal microscopy, either near the glass (ventral) or about 5–10 μm above (medial). Concave white arrowheads point at examples of Rock1/2 accumulation. (D, E, H, I) Localization relative to the cell cortex and to Vinculin-Cherry labeled cell-matrix adhesive structures (red arrowheads). (F, G, J, K) Localization relative to cell–cell contacts, marked by cadherin-dTomato (red arrows). (D, E) In the ectoderm, Rock1 and 2 have both a cortical localization. Levels are low on the ventral side inside the adhesive ring, but stronger outside of the ring, particularly for Rock2. (F, G)

Levels are very low at cell–cell contacts. (H, I) In the ventral face of mesoderm cells, Rock1 tends to be enriched in the central part, Rock2 at the periphery of the protrusions. Both are low at FAs. They both accumulate at the cortex along cell free edges (medial planes). (J, K) Levels are low at cell–cell contacts. y: autofluorescence of yolk platelets, abundant in mesoderm cells. FA, focal adhesion; MLC, myosin light chain; mYFP, membrane-targeted YFP; Rock, Rho-kinases.
(PDF)

**S3 Fig.** (Related to Fig 2) (A, B) Area expansion for single cells after treatment with Rock inhibitors Y27632 (50 μM) and H1125 (1 μM). Average and SD of 107 cells (A) and 34 cells (B). (C) Changes in vinculin distribution. Images from a time-lapse movie of a small group of 3 cells expressing Vinculin-Cherry, treated at time = 0 with Y27632. Filled arrowheads: ring-like adhesion; concave arrowheads: FAs. Scale bars: 10 μm. (D, E) Opposite effects of Rock and MLCK inhibition on cell adhesion. Ectoderm and mesoderm adhesion to FN or cadherin was measured after treatment with Rock inhibitors Y27632 (Y, 50 μM), H1125 (H, 1 μM), or the MLCK inhibitor ML7. Five experiments, a total of 1,000–2,000 cells/conditions. Statistical comparison to control ectoderm or mesoderm, comparing the % adherent cells/experiment, pairwise 2-sided Student $t$ test. Refer to S1 Data. FA, focal adhesion; FN, fibronectin; SD, standard deviation.
(PDF)

**S4 Fig.** (Related to Fig 4) (A–D) Rescue of Rnd1MO and ShiMO spreading and migration phenotypes. Four-cell stage embryos were injected in the dorsal side with COMO, RndMO, RndMO + YFP-Rnd1 mRNA (rescue), ShiMO, or ShiMO + YFP-Shirin mRNA (rescue). Dissociated mesoderm cells were plated on FN and time-lapse movies were recorded. The fourth condition represents RndMO or ShiMO cells treated with 50μM Y27632 Rock inhibitor (Y). Statistical comparions: 1-way ANOVA followed by Tukey HSD post hoc test. Red asterisks: Comparison to COMO. (E) Migration speed for different cell morphology categories. Analysis of data from Fig 4I. Red asterisks: comparison to COMO. One-way ANOVA followed by Tukey HSD post hoc test. Refer to S1 Data. ANOVA, analysis of variance; FN, fibronectin; HSD, honestly significant difference.
(PDF)

**S5 Fig.** (Related to Fig 5) (A) Morphometry of Rnd1 and Shirin induced spreading of ectoderm cells. The diagrams illustrate typical cell shapes. Corresponding images can be found in main Fig 5A–5E. These shapes were analyzed based on the following parameters: (A') Area of the ventral contact surface (red in the schemes in A). (A") Circularity of the ventral surface, which depends both on the roundness and regularity/convolution of the shape. (A''') Ratio between the ventral area and the maximal cell area, calculated from maximal z projections. Refer to S1 Data. Blebs were excluded from measurements. Rnd1- and Shirin-expressing cells were here subdivided in 2 categorizes, low and high expression, based on the YFP fluorescence intensity. Note that these 2 categories overlap but are not equivalent Rnd1 expression levels had no significant impact on any parameter. Shirin expression had no effect on contact surface area, but high levels stimulated formation of convoluted protrusions (lower circularity) but decreased ventral/max area, reflecting the fact that many of them rounded up (fourth cell shape in panel A; see main Fig 5E). (B) Distribution of speed for ectoderm cells expressing Rnd1 or Shirin, compared to wild-type ectoderm and mesoderm. Brackets: range of high speed, comparable to mesoderm, achieved mainly by Rnd1-expressing cells. Refer to S1 Data.
(PDF)

**S6 Fig.** (Related to Fig 6) Effect of Rnd1 and Shirin ectopic expression. (A) Loosening of ectoderm tissue upon expression of Shirin. Immunostained section of a YFP-Shirin expressing embryo showing a loosely organized ectoderm, characterized by the presence of large intercellular spaces (asterisks) and heterogenous β-catenin signal, weak signal along membranes except for strong local concentrations (arrowheads). Scale bar, 10 μm. (B–E) Cell orientation. (B–D) The main axis of deep ectoderm cells (double arrows) tend to orient roughly perpendicular to the inner surface of the tissue (dashed line). Rnd1-expressing cells show variable orientation. Shirin-expressing cells align parallel to the surface. Scale bars, 10 μm. (E) Quantification of the angle between the cell axis and the tissue interface. Numbers into brackets correspond to number of cells/embryos/experiments. Refer to S1 Data. (F–I) Analysis of β-catenin (green) and pMLC (red) in ectoderm explants. (F) Diagram, section of a control ectoderm explant (scale bar, 50 μm) and quantification. Statistical comparison using 1-way ANOVA followed by Tukey HSD post hoc test. (G–I) Examples of ectoderm explants. (G) β-catenin and pMLC signal along cell edges is highest in control (arrows). H,I) Explants expressing Rnd1 or Shirin. β-catenin tends to accumulate at cell vertices (concave arrowheads). pMLC levels are lower except for some cells (I, asterisks) that have rounded up and display high pMLC throughout the cell. Little to no β-catenin is seen between the round cells. Scale bars, 20 μm. (H) Effect of Rnd1 and Shirin expression on phosphorylation of MYPT. Dissected ectoderm tissues were analyzed by western blot. GAPDH was used as loading control, and the pMYPT signal was expressed as relative ratio, normalized to ectoderm control set to 1.0. Three independent experiments, statistical analysis using 1 sample, 2-sided *t* test. Refer to S1 Data. ANOVA, analysis of variance; HSD, honestly significant difference; MYPT, myosin light chain phosphatase; pMLC, phosphorylated MLC.
(PDF)

**S7 Fig.** (Related to Fig 9) (A) Example of ectoderm explant showing late partial spreading, which is only observed beyond the 120 minutes. (B) Examples of traces for single explants, illustrating the irregular expansion of mesoderm explants interrupted by retractions. In contrast, expansion of Y27632-treated mesoderm is smooth. (C) Quantification of average triangle size at the initiation of spreading (30 minutes) and the end of the time lapse (170 minutes). Refer to S1 Data.
(PDF)

**S8 Fig. (Related to Fig 9) Knockdown of Rnd1 or Shirin affect collective properties of mesoderm tissue explants.** Analysis of spreading, dispersion, and intercalation of mesodermal explants under various conditions was performed as for experiments presented in Fig 9. (A–D) Control mesoderm, mesoderm treated with Y27632, and mesoderm from embryos injected with Rnd1 MO or Shirin MO. Red arrowheads in A and D indicate areas of large-scale retractions (compare 85 and 170 minutes). Scale bar: 100 μm. (E) Average time course curves with SD for the various experimental conditions. (E') Corresponding relative spreading after 60 minutes and 170 minutes. (F–K) Delaunay triangulation of nuclei in order to measure cell dispersion. (F–I) Representative plots of triangulated nuclei after 170 minutes of imaging. X and Y labels mark the coordinates in μm, and the color-coded scale bar indicates the area of the triangles in $μm^2$. (J) Quantification of average triangle size at the initiation of spreading (30 minutes) and the end of the time lapse. (K) Quantification of the relative change in triangle size over time calculated by dividing the average triangle area at 170 minutes by that at 30 minutes. (L) Quantification of intercalation calculated by dividing number of nuclei at the ventral surface at 170 minutes by the number at 30 minutes. Statistical comparisons: 1-way ANOVA followed by Tukey HSD post hoc test. Refer to S1 Data. ANOVA, analysis of variance; HSD,

honestly significant difference; SD, standard deviation.
(PDF)

**S1 Appendix. Calculation of relative tensions and adhesiveness for cell doublets.**
(PDF)

**S1 Data. Excel file that includes all individual numerical data in the different figure panels.**
(XLSX)

**S1 Raw Images. Full gel images of western blots of panel H, S6 Fig.**
(PDF)

**S2 Raw Images. Full gel images of western blots of panel H, S6 Fig.**
(PDF)

**S1 Movie. Related to Fig 1.** Live imaging of ectoderm cell plated on fibronectin. Red: membrane-Cherry, green: paxillin-YFP. Left: ventral plane, right: maximal z projection. One frame every 5 minutes for 50 minutes. 60× objective. Scale bar: 10 μm.
(AVI)

**S2 Movie. Related to Fig 1.** Live imaging of mesoderm cell plated on fibronectin. Red: membrane-Cherry, green: paxillin-YFP. Ventral plane. One frame every 5 minutes for 50 minutes. 60× objective. Scale bar: 10 μm.
(AVI)

**S3 Movie. Related to Fig 2.** Ectoderm cell treated with 50-μM Y27632. Red: Vinculin-Cherry, green: membrane YFP. One frame every 4.5 minutes for 45 minutes. 60× objective. Scale bar: 10 μm.
(AVI)

**S4 Movie. Related to Fig 2.** Group of ectoderm cells treated with 50-μM Y27632. Left: ventral plane. Right: maximal z projection. One frame every 4.5 minutes for 45 minutes. Red: Vinculin-Cherry, green: membrane YFP. 60× objective. Scale bar: 20 μm.
(AVI)

**S5 Movie. Related to Fig 4.** Migration of single cells on fibronectin. Examples of control ectoderm, control mesoderm, mesoderm + Rnd1MO, + ShiMO, + ShiMO + 75pg Shirin mRNA (rescue), and + ShiMO + Y27632 treatment. One frame every 2.5 minutes. Bright field, 10× objective.
(AVI)

**S6 Movie. Related to Fig 5.** Examples of migration of ectoderm cells ectopically expressing Rnd1 (250 pg mRNA) or Shirin (7 5pg mRNA). One frame every 2.5 minutes. Bright field, 10× objective.
(AVI)

**S7 Movie. Related to Fig 9.** Wild-type ectoderm explant on fibronectin. Merge of bright field and blue fluorescence of Hoechst-labeled nuclei.
(AVI)

**S8 Movie. Related to Fig 9.** Ectoderm treated with Y27632.
(AVI)

**S9 Movie. Related to Fig 9.** Ectoderm expressing Rnd1 (250 pg mRNA).
(AVI)

**S10 Movie. Related to Fig 9.** Ectoderm expressing Shirin (75 pg mRNA).
(AVI)

**S11 Movie. Related to Fig 9.** Control mesoderm.
(AVI)

**S12 Movie. Related to Fig 9.** Mesoderm treated with Y27632.
(AVI)

**S13 Movie. Related to Fig 9.** Mesoderm + Rnd1MO.
(AVI)

**S14 Movie. Related to Fig 9.** Mesoderm + ShiMO.
(AVI)

**S15 Movie. Related to Fig 10.** Pipette aspiration of control ectoderm explant.
(AVI)

**S16 Movie. Related to Fig 10.** Pipette aspiration of control mesoderm explant.
(AVI)

## Acknowledgments

F.F. dedicates this study to Ray Keller, intrepid adventurer of morphogenesis and inspiring master. We thank Drs. C. Niessen and B. Gumbiner for generous gift of the C-cadherin-Fc-expressing CHO cell line and Dr. K. Cho, F. Spagnoli, A. Brivanlou, N. Kinoshita, and S. Yonemura for plasmids. We are deeply grateful to Drs. Karine Guevorkian and David Gonzales-Rodriquez for invaluable help with setting and interpreting MPA experiments. We thank the Montpellier Rio Imaging platform for technical support.

## Author Contributions

**Conceptualization:** Leily Kashkooli, David Rozema, François Fagotto.

**Formal analysis:** Leily Kashkooli, David Rozema, François Fagotto.

**Funding acquisition:** Paul Lasko, François Fagotto.

**Investigation:** Leily Kashkooli, David Rozema, Lina Espejo-Ramirez, François Fagotto.

**Methodology:** Leily Kashkooli, David Rozema, Lina Espejo-Ramirez, François Fagotto.

**Supervision:** Paul Lasko, François Fagotto.

**Writing – original draft:** Leily Kashkooli, David Rozema, Paul Lasko, François Fagotto.

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
