## [Editor Report · Decision Letter 0]

5 Mar 2020

Dear Dr Fagotto, 

Thank you for submitting your manuscript entitled "Ectoderm to mesoderm transition by downregulation of actomyosin contractility" for consideration as a Research Article by PLOS Biology.

Your manuscript has now been evaluated by the PLOS Biology editorial staff as well as by an academic editor with relevant expertise and I am writing to let you know that we would like to send your submission out for external peer review.

Please re-submit your manuscript within two working days, i.e. by Mar 09 2020 11:59PM.

Kind regards,

Ines

--

Ines Alvarez-Garcia, PhD

Senior Editor

PLOS Biology

Carlyle House, Carlyle Road

Cambridge, CB4 3DN

+44 1223–442810

---

## [Decision Letter · Decision Letter 1]

12 May 2020

Dear Dr Fagotto,

Thank you very much for submitting your manuscript "Ectoderm to mesoderm transition by downregulation of actomyosin contractility" for consideration as a Research Article at PLOS Biology. Thank you also for your patience as we completed our editorial process, and please accept my apologies for the delay in providing you with our decision. Your manuscript has been evaluated by the PLOS Biology editors, an Academic Editor with relevant expertise, and by three independent reviewers.

You will see that the reviewers agree that the study addresses an interesting question and contains some exciting results, but they also raise several issues that need to be addressed before we can consider the manuscript for publication. While Reviewers 1 and 2 request several clarifications, Reviewer 3 raises some concerns regarding the use of explants and the mechanistic insights. After consulting with the academic editor, we do feel that you should specifically address Reviewer 3's Points 1 (discussing the in vitro limitations at length) and 5 (the role of Rho and Shirin on cell-cell junction formation/maintenance). Please address the rest with text changes or re-analysis of some of the data.

In light of the reviews (attached below), we will not be able to accept the current version of the manuscript, but we would welcome re-submission of a revised version that takes into account the reviewers' comments. We cannot make any decision about publication until we have seen the revised manuscript and your response to the reviewers' comments. Your revised manuscript is also likely to be sent for further evaluation by the reviewers.

We expect to receive your revised manuscript within 3 months. 

Please email us (plosbiology@plos.org) if you have any questions or concerns, or would like to request an extension due to the closure of the labs because of the COVID-19 pandemia. At this stage, your manuscript remains formally under active consideration at our journal; please notify us by email if you do not intend to submit a revision so that we may end consideration of the manuscript at PLOS Biology.

**IMPORTANT - SUBMITTING YOUR REVISION**

*Re-submission Checklist*

*Published Peer Review*

*PLOS Data Policy*

*Blot and Gel Data Policy*

Sincerely,

Ines

--

Ines Alvarez-Garcia, PhD

Senior Editor

PLOS Biology

Carlyle House, Carlyle Road

Cambridge, CB4 3DN

+44 1223–442810

Reviewers comments

Rev. 1:

In this paper, Kashkooli et. al. provide compelling evidence that the ectoderm to mesoderm transition in gastrulating Xenopus embryos involves downregulation of actomyosin contractility. Furthermore, they show this reduction in contractility is driven by two negative regulators of RhoA, namely Rnd1 and Shirin. Some experiments were performed in intact embryos, but most of the experiments assayed the morphology and migratory behavior of explants or isolated cells when plated on sparse FN in vitro. Mesoderm and ectoderm show distinct morphologies: ectoderm are round and compact with a stable ring of adhesions and blebs, while mesoderm are spread, protrusive, with dynamic adhesions and are more migratory. Acute treatment with Rock inhibitors causes the ectoderm to behave more similar to mesoderm, except that it migrates slower. Rnd1 and Shirin are more highly expressed in the mesoderm. Morpholino knockdown of either resulted in gastrulation defects, with a block in mesoderm involution. In vitro, mesoderm KD for Rnd1 or Shirin look like ectoderm, and the phenotype can be rescued by Rock inhibition. However, differences in cell-matrix adhesion patterns and migration rates were observed between Rnd1 and Shirin MO. Overexpression of Rnd1 or Shirin in ectoderm cells rendered them mesoderm-like. Rnd1 localizes to the cortex and Shirin to cell-matrix adhesion and protrusions. Micropipette aspiration assays documented the changes to the physical properties of the mesoderm and ectoderm tissues after contractility inhibition by a drug or Rnd1/Shirin expression in ectoderm.

Overall, I found the paper to be clear, well executed and controlled, and intriguing. I did not find any major problem that should preclude it from being accepted for publication. Below is a list of minor issues I recommend addressing before the actual publication:

1. Stars of statistical significance need to be on a comparison, not on a single column.

2. Figure 1, why is D' not the same cells as D?

3. When describing the results of the adhesion assay by rotation the authors write about ectoderm binding to FN that they "adhered almost as efficiently as mesoderm cells" and when describing their adhesion to cadherin they write "Ectoderm cells showed significantly higher cadherin adhesion than mesoderm cells". However, in both cases the difference is modest and yet statistically significant and so should be described the same way.

4. Fig.1C', yellow concave arrows - there are no yellow concave arrows in 1C'. Maybe meant yellow arrows in 1C?

5. Fig.1D', white and yellow concave arrows - should be fig1D?

6. The authors write: "In all these aspects, Rock inhibition appeared sufficient to induce a dramatic transformation of ectoderm cells into mesoderm- like cells." - It should be noted, as seen in Fig2G, that the migration speed is still far from mesoderm.

7. Regarding the experiments with whole embryos the authors write: "While there were differences between the detailed phenotypes of the two knockdowns, their analysis was of little informative value toward understanding the underlying mechanisms." - why? What about the double KD?

8. In figure 3 in the graphs F, G there is a condition of Rnd1MO + ShiMO - this condition is not mentioned in the results text.

9. Shirin has strong effect on cortical tension and cell spreading while rnd1 has mild effect. However, Rnd1 expression in the mesoderm is much stronger relative to ectoderm compared to shirin. The authors could discuss why this might be the case.

10. "From the comparison of the effects of Rnd1 and Shirin on tension (Fig.5J) and on cell spreading (Fig.4G), we can extrapolate that a ~20% reduction in cortical tension may be sufficient to allow ectoderm cells to start to elongate and spread." - Not clear how this extrapolation was done.

11. "For this purpose, we dissected tissue explants, let them heal," - what does that mean? How long does the healing process take?

Rev. 2:

This manuscript investigates how the differences in morphology and motility between mesoderm and ectoderm are controlled in Xenopus. The central hypothesis of this paper is that these differences arises from differences in the expression of regulators of Rho-kinase. This is an important topic as the control of mechanics in the different germ layers at the cellular level will play a key role in tissue morphogenesis. It is also interesting that mechanical differences are controlled by only few proteins. The experiments are interesting and generally well quantified using a muli-disciplinary approach. Overall, I have only few issues with the data and most can be resolved with image analysis. Following revisions, I recommend publication.

Major issues:

-page 9, paragraph 1: the authors state that shirin expressing explants become looser with many cells migrating away whereas rnd1 explants spread while remaining compact. This is an important result and it would be good to quantify this more carefully. Maybe by looking at a triangulation of the cell centroids and the average area of the triangles over time as others have done for neural crest? Or by quantifying the distribution of numbers of neighbours over time?

-Do the authors have western blots showing that in Rnd1 or Shirin expression in ectoderm explants decreases the level of RhoGTP or pROCK compared to control explants? Given the current circumstances, if the authors have not acquired this data, it is ok to ignore this comment. But if they do have it, it would be nice to add it because it will make the role of Rnd1 and Shirin as regulators of Rho-ROCK signalling during morphogenesis more convincing.

-page 6, last paragraph: the authors state that shirin expression caused cells to overspread. The authors should quantify cell spreading. This would be a good complement to the morphological classification in panel G.

-Could the authors give more details about the calculation of the various mechanical parameters they extract from the doublet experiments? An appendix would be helpful for the readers and make the paper self contained. It would also allow me to understand exactly what the authors have done, which I find a bit difficult to judge currently.

Minor issues:

-In some of the figures, the control is the mesoderm (e.g. Fig 3), in others it is the ectoderm (e.g. Fig 4). The authors have tried to colour code this but there are so many colours involved in the graphs that it is difficult to remember what colour represents what. On the legend, they just write COMO or ctrl. Could the authors also add the name of the germ layer (Como meso or ctrl ecto)?

-In the discussion, paragraph 3, the solid to liquid transition for biological materials is not an all or nothing transition. There is a whole spectrum of properties that are more solid-like or more liquid-like.

Rev. 3:

Ectoderm to mesoderm transition is a fundamental process in embryogenesis, leading to the transition of resting epithelium to moving mesodermal cells. Kashkooli and co-workers here provide compelling evidence that the inhibition of the Rho/ROCK axis is sufficient to induce a mesenchymal-like motile behavior in isolated and grouped epithelial cells. Using expression analysis and morpholino-based interference, the authors show that the Rho inhibitors Rnd1 and Shirin are both upregulated in mesodermal cells and that downregulation of both reverts the phenotypic switch and adhesion to fibronectin and cadherin ligand in vitro, and perturbs gastrulation in vivo. Rnd1, but not Shirin, was required for focal adhesion formation, whereas Shirin preferentially supported speed gain. Thus, Rnd1 and Shirin, by regulating Rho, critically contribute to ectoderm-to-mesoderm transition.

The concept that lowering Rho activity induces epithelial cell kinetics as the onset for mesoderm migration is compelling, and that Rnd1 and Shirin control Rho in this process is functionally worked out well. The multicellular explant culture delivers a reliable model for quantifying cell kinetics and dispersion. However, by which mechanism Rho regulation initiates this transition, directly vial actomyosin regulation or via affecting cell-cell junctions, and how Rnd1/Shirin differentially regulate this complex process, independently or in series, remain unclear.

In addition, mechanistic key claims are based on in vitro findings, using isolated cells monitored in 2D culture systems in short-term culture (few hours), whereas in vivo validation was only performed for global endpoint analysis. It remains unclear how the identified phenotypes and derived conclusions hold up when more thoroughly compared to in vivo analyses. Lastly, while the data show that Rnd1/Shirin are required to initiate gastrulation, it is not clear whether they are sufficient to trigger gastrulation in vivo.

1. In vivo, both ectodermal and mesenchymal states retain cell-cell junctions, enabling either cohesive epithelial barrier formation or collective invagination and mesoderm invasion. However, most in vitro results were derived using fully isolated cells, which disregards that cell-cell junction regulation is involved in gastrulation. How is the initiation of gastrulation represented by the cell-substrate (FN) interaction of isolated cells? Is indeed a focal adhesion formation required for initiating or maintaining gastrulation? Can the morphological and adhesion features, and the speed data more precisely be linked to the in vivo phenotype? The significance of the mechanistic conclusions could be strengthened by performing in vitro and time-lapse in vivo analyses side-by-side to validate at least key points on cell function for the multicellular process in vivo.

2. The roles of Rnd1 and Shirin were tested using morpholino-based downregulation, and this resulted in defective cell-substrate interaction and migration in vitro. However, it is not clear whether both regulators mediate gastrulation, and whether their induction suffices to induce mesoderm migration. Is ectopic expression of YFP-Rnd1 or YFP-Shirin sufficient to induce gastrulation in multicellular ectodermal explants and in vivo?

3. How much Rho silencing is required to induce epithelial dynamics, and how much Rho activity is required to maintain mesenchymal movement? There is a potential fine line between both requirements. How effective was the Rho/ROCK inhibition in either setting, including knockdown experiments? Biochemical (if feasible) or in situ analysis of the ROCK inhibitor effects should be provided, such as pMLC/MLC stainings after molecular intervention in vitro and in vivo. Which level of Rho/ROCK activity is required to maintain mesoderm migration?

4. Descriptors/measures of ectodermal-to-mesoderm transition are not well described and consistently used throughout the manuscript. While gain of invasiveness (migration speed, cell area) and single cell dispersion (cohesion) are features of the mesodermal state, these parameters are not consistently quantified after molecular interference. As a consequence, we are left uncertain which overlapping and distinct roles of Rnd1 and Shirin on cell-cell cohesiveness and substrate adhesion underlying the transition in individualized cells, multicellular explants and in vivo are.

5. Rho silencing along cell-cell junctions is an important mechanism to support cell-cell cohesion, so their interventions are expected to affect both cell-substrate and cell-cell adhesions. The authors focus on actomyosin contractility and substrate adhesion, but neglect the role of Rho on cell-cell junction regulation. How is the viscosity regulation shown in Fig. 8 linked to adherens junction stability and cortical actin organization along adherens junctions? How are cell-cell interactions perturbed by Rnd1 and Shirin regulation and how critical are they in permitting ectoderm-to-mesoderm transition?

Additional points

Fig 1G: The authors argue that the ectodermal cells do not migrate, while the data on 2D show a range of migration speed up to 1 um/min, indicating moderate-speed behavior. The authors should phrase these findings with greater precision. Is this baseline level of epithelial movement in vitro consistent with the relative immobility in vivo?

Fig. 1E: The confocal images indicate that vinculin intensity is rather similar between the ectoderm and mesoderm, but the peaks may differ. These differences should be reflected in their image analysis.

All functional studies and analysis after ROCK inhibitor treatment lack control treatments with solvent (presumably DMSO). DMSO may affect cell viability and function, therefore appropriate control data should be provided.

Fig 2. ROCK inhibition increases the migration speed of ectoderm cells, however, not to the extent of migration speed of the mesoderm. Can the authors explain this incomplete functional transition?

Fig 3C-D: The cells show surprisingly little cortical actin. Are they healthy? Viability analysis should be performed to exclude toxic effects after morpholino knockdown.

The role of Rnd1 and Shirin in vivo is currently in supplementary figure S3, but could be presented as main data.

Fig 4A-C: The cells' appearance seems to differ from other datasets. Are the dot-like structures indeed focal adhesion or rather vacuoles? Toxic effects?

Fig 6: The dot-like cad-tom structures are not explained or mentioned in text. The co-localization of Rnd1 and Shirin with vinculin should be quantified over a larger number of cells.

In the ectoderm ShirinMO experiment, a higher number of single cells was released, compared to Rnd1MO expression, indicating an effect on cell-cell junctions. By immunofluorescence (Fig. 6), particularly Rnd1 locates at cell-cell contacts. How are these data connected to the expected roles of Rnd1 and Shirin at cell-cell junctions?

Movie 5 and 6: It is unclear what each subpanel shows, as the labels for experimental conditions are missing. Figures: Several y axis labels are lacking, e.g. Fig. 1E,H, Fig. 2E,F,H

Statistical analysis is unclear. In several figures it is unclear which experimental condition was compared to which control, e.g. in Fig 3 and Fig S4. Lines should be included in the figure, and definitions for p values are missing in all legends. The figure legend should further include information about analysis and number of independent experiments.

---

## [Decision Letter · Decision Letter 2]

17 Nov 2020

Dear Francois,

Thank you very much for submitting a revised version of your manuscript entitled "Ectoderm to mesoderm transition by downregulation of actomyosin contractility" for consideration as a Research Article at PLOS Biology. Once again, please accept my apologies for the delay in providing you with our decision. This revised version of your manuscript has been evaluated by the PLOS Biology editors, by the Academic Editor and by two of the original reviewers.

The reviews are attached below. You will see that the reviewers are mostly satisfied with the revision, but raise some remaining points that should be addressed, in particular Reviewer 2's comment regarding a potential error in one of the calculations (relationship (4) - see also the attachment for full details). Thus we are pleased to offer you the opportunity to address the points raised by the reviewers in a revised version that we anticipate should not take you very long. We will then assess your revised manuscript and your response to the reviewers' comments.

We expect to receive your revised manuscript within 1 month.

**IMPORTANT - SUBMITTING YOUR REVISION**

*Resubmission Checklist*

*Published Peer Review*

*PLOS Data Policy*

*Blot and Gel Data Policy*

Sincerely,

Ines

--

Ines Alvarez-Garcia, PhD

Senior Editor,

PLOS Biology

Reviewers’ comments

Rev. 2:

The authors have addressed my comments satisfactorily and in my view the article is much improved by the additional experiments. Before publication, I have one remaining major issue that must be addressed. This is the relationship giving TAB/CtB (relationship (4)).

If I take the expressions (1) and (2) provided in the appendix and try to work out the relationships, I find something different from relationship (4).

Expression (2) can be rewritten to give CtA as a function of TAB and CtB:

(see attachment)

Which is different from (4) in the appendix. My guess is that (4) is in fact TAB/CtA. There is also an alpha that appears in their relationship, which I guess is a typo.

The authors should recheck their appendix and reevaluate everything that derives from (4). It would also be much clearer if they used some equation editing software – like Latex or the word equation editor. Relationship (6) is very difficult to read and could be simplified by noting that (TAB/CtA)/(TAB/CtB) is in fact CtB/CtA.

Rev. 3:

The authors introduce the term "ectoderm to mesoderm transition", or "mesoderm transition" to label the fluidization of an epithelium and transition to collective movement. While appropriate for their system, this label may be misleading when epithelia become activated, initiate collective movement, but do not reach mesoderm position or function, such as during wound healing of glandular sprouting. Thus, the authors should reconsider their terminology, or justify the introduction of new terms by detailing the lack of already existing nomenclature. Additional discussion on peculiarities of the in vitro/in vivo comparison aid the general appreciation of methodological choices and limitations.

Lastly, new triangulation data (suggested by referee 1) add depth to the morphological transition analysis. In summary, I applaud to this comprehensive revision.

---

## [Editor Report · Decision Letter 3]

30 Nov 2020

Dear Francois,

Thank you for submitting your revised Research Article entitled "Ectoderm to mesoderm transition by downregulation of actomyosin contractility" for publication in PLOS Biology. I have now obtained advice from the Academic Editor and have discussed the revision with the team of editors. 

We're delighted to let you know that we're now editorially satisfied with your manuscript and we will start the process to prepare it for production.

Before we can formally accept your paper and consider it "in press", we need to ensure that your article conforms to our guidelines. A member of our team will be in touch shortly with a set of requests. As we can't proceed until these requirements are met, your swift response will help prevent delays to publication. Please also make sure to address the data and other policy-related requests noted at the end of this email.

To submit your revision, please go to https://www.editorialmanager.com/pbiology/ and log in as an Author. Click the link labelled 'Submissions Needing Revision' to find your submission record.

*Copyediting*

*Published Peer Review History*

*Early Version*

Best wishes,

Ines

--

Ines Alvarez-Garcia, PhD,

Senior Editor,

ialvarez-garcia@plos.org,

PLOS Biology

Fig. 1E, G, H; Fig. 2D, E; Fig. 3A, F, F’; Fig. 4D-H; Fig. 5F, G, I-K; Fig. 6A’, C’; Fig. 7C, H; Fig. 8J-L; Fig. 8G, G’, H, H’, I-N; Fig. 10A’, B’, D-I; Fig. S1B; Fig. S2C; Fig. S3A, B, D, E; Fig. S4A-E; Fig. S5A’, A’’, A’’’, B; Fig. S6E, F’, H; Fig. S7B, C and Fig. S8E, E’, F-L

For manuscripts submitted on or after 1st July 2019, we require the original, uncropped and minimally adjusted images supporting all blot and gel results reported in an article's figures or Supporting Information files. We will require these files before a manuscript can be accepted so please prepare and upload them now. Please carefully read our guidelines for how to prepare and upload this data: https://journals.plos.org/plosbiology/s/figures#loc-blot-and-gel-reporting-requirements.

---

## [Editor Report · Decision Letter 4]

29 Dec 2020

Dear Dr. Fagotto,

I am writing concerning your manuscript submitted to PLOS Biology, entitled “Ectoderm to mesoderm transition by downregulation of actomyosin contractility.”

We have now completed our final technical checks and have approved your submission for publication. You will shortly receive a letter of formal acceptance from the editor.

Kind regards,

PLOS Biology